# Grassland ecological compensation policy in China improves grassland quality and increases herders' income

Lingling Hou[1]✉, Fang Xia[2], Qihui Chen [3]✉, Jikun Huang[1], Yong He [4]✉, Nathan Rose [5] & Scott Rozelle[5]

Many countries have undertaken large and high-profile payment-for-ecosystem-services (PES) programs to sustain the use of their natural resources. Nevertheless, few studies have comprehensively examined the impacts of existing PES programs. Grassland Ecological Compensation Policy (GECP) is one of the few pastorally focused PES programs with large investments and long duration, which aim to improve grassland quality and increase herder income. Here we present empirical evidence of the effects of GECP on grassland quality and herder income. Through a thorough and in-depth econometric analysis of remote sensing and household survey data, we find that, although GECP improves grassland quality (albeit to only a small extent) and has a large positive effect on income, it exacerbates existing income inequality among herders within their local communities. The analysis demonstrates that the program has induced herders to change their livestock production behavior. Heterogeneity analysis emphasizes the importance of making sure the programs are flexible and are adapted to local resource circumstances.

[1] China Center for Agricultural Policy, School of Advanced Agricultural Sciences, Peking University, Beijing, China. [2] Research Institute for Global Value Chains, University of International Business and Economics, Beijing, China. [3] Beijing Food Safety Policy & Strategy Research Base, College of Economics and Management, China Agricultural University, Beijing, China. [4] Institute of Environment and Sustainable Development in Agriculture, Chinese Academy of Agricultural Sciences, Beijing, China. [5] The Freeman Spogli Institute for International Studies, Stanford University, Stanford, CA, USA. ✉email: llhou. ccap@pku.edu.cn; chen1006@umn.edu; heyong01@caas.cn

Ecosystem overexploitation remains a critical global environmental problem, and policymakers across the globe need to take action to protect the environment from overexploitation while simultaneously increasing (or at least protecting) the welfare of their country's population[1]. Payment-for-ecosystem-services (PES) programs have emerged as a potential way to address both of these needs. PES programs work through voluntary transactions, whereby people who benefit from environmental services pay those who provide such services, essentially creating a market for conservation[2].

Over the past few decades, many countries have undertaken large and high-profile PES programs to sustain the use of their natural resources, especially farmlands and forests[3]. The Conservation Reserve Program in the United States, the Agri-Environmental Scheme in the European Union, and the Grain-for-Green Project in China are well-known examples of PES programs[4–10]. Although market-based PES programs are gaining in popularity, the number of counterfactual-based impact evaluations of these programs has remained too small to draw valid, generalizable conclusions about the effects of such programs[11]. Moreover, most of the current PES programs focus on farmland or woodland preservation, and a review by Adhikari and Agrawal[12] of 26 different PES programs found none that concerned grassland and pastoralists.

The lack of PES programs on grassland across the globe appears to be a major omission, as grassland ecosystems are as important and vulnerable as are other ecosystems. Grasslands are a major part of the global ecosystem, covering 20–40% of the land surface, based on various measurements[13]. Nearly half of all grasslands worldwide, however, are experiencing degradation[14]. In addition to being essential ecologically vital, grasslands also are economically vital. Globally, about 30% of the supply of meat is produced from grasslands[15]. In China alone, nearly 18 million herdsmen in pastoral or semi-pastoral areas live on grasslands, with grazing livestock as their most important source of income[16]. Despite the importance of grasslands, however, few countries have created PES programs that protect their grasslands and the pastoralists whose livelihoods rely on them.

The Grassland Ecological Compensation Policy (GECP) in China is, thus, a rare example of a pastoralist-focused PES program. At the time of this writing, it is the world's largest PES grassland conservation program in terms of area, the number of participants, and total monetary transfers. With the dual goals to restore grassland ecosystems and raise herder income, the first five-year program, GECP-I, was implemented in eight grassland-rich provinces between 2011 and 2015[17]. During this period, the central government invested 77.4 billion RMB (over 10 billion US dollars) to implement the program. Most of the program funding was paid directly to herders to compensate for reductions in grazing intensity or the cessation of grazing activities.

Without any formal evaluation of the first GECP-I program, a second five-year program, GECP-II, was launched in 2016[18]. GECP-II included the original eight provinces plus five additional provinces and was supported with an even larger budget of 93.8 billion RMB (nearly 15 billion US dollars)[19]. In recent years, GECP payments have been automatically transferred directly to the personal bank accounts of all herders who have grassland-use rights within the project areas; that is, payments are made without an application or formal herder performance evaluation. A household's total GECP income depends on the acreage on their grassland certificate, as total GECP payment is calculated by multiplying the total certified area managed by a household by a regulatory standard (yuan/ha). This implies that the more certified grassland a herder has, the larger the GECP payment the herder receives. The central government sets a uniform national payment standard, but each province is allowed to set county-specific payment standards according to its historical record of grassland quality. A county with high baseline grassland quality will be assigned a higher payment standard to compensate for higher levels of lost income from the livestock sector due to GECP implementation.

Given China's large investment in the program and the length of time (10 years) since the launch of the program, it is surprising that only two studies have empirically examined the impacts of GECP. Yet even these two existing studies are limited in scope. Hu et al.[20] analyzed how the program affected herder behavior (e.g., the total number of livestock in stock) without examining either of the two stated policy objectives. Liu et al.[21] investigated only one of two policy objectives, evaluating GECP's effect on grassland quality but not its impact on herder income. To the best of our knowledge, thus far, no studies have analyzed the impacts of GECP on either herder income or equity among herders within their communities.

The studies of Hu et al.[20] and Liu et al.[21] differ also in terms of their findings. Whereas Liu et al. concluded that GECP significantly improved grassland quality in Inner Mongolia by about 30%, Hu et al. demonstrated that GECP had little impact on livestock production in the same region. One reason for this divergence in findings may be that Liu et al.'s study did not include a proper control group.

In this work, we show that although the GECP slightly improves grassland quality and has a large positive effect on income, the program exacerbates existing income inequality between herders within their local communities. Our mechanism analysis shows that, in response to the program, herders reduce livestock production. However, further increasing the payment standard without augmenting other aspects of implementation would not further reduce livestock production, but could encourage herders to enlarge farm size by renting in grassland and increase supplementary feedings as responses. Our heterogeneity analysis on ecological impacts indicates that investing in rural roads, expanding farm size, and enhancing herder monitoring can help to enhance ecological gains.

## Results

**Impact on grassland quality.** Overall grassland quality, as measured by the Normalized Difference Vegetation Index (NDVI), slightly improved after GECP implementation, but there was significant variation across regions. On average, NDVI increased by 1.2% in GECP regions from the pre-program period (2006–2010) to the program period (2011–2015) (Supplementary Table 1). Figure 1 shows the change in grassland quality by displaying NDVI measures before GECP implementation (Fig. 1a), NDVI measures after implementation (Fig. 1b), and the difference in NDVI between the two periods (Fig. 1c). Figure 1 also indicates that NDVI increased in some areas and decreased in other areas.

Using a county-level DID (difference-in-differences) method, which better controls for confounding factors, we found that GECP's impact on grassland quality is positive and statistically significant but small in magnitude. In this DID setup, the treatment group includes all counties in the five North and Northwestern program provinces that were covered in GECP-I in 2011–2015 (i.e., Xinjiang, Qinghai, Gansu, Ningxia, and Inner Mongolia). The control group includes all counties in the five North and Northeastern provinces that were not covered by GECP-I until 2016 (i.e., Shanxi, Hebei, Liaoning, Jilin, and Heilongjiang). The pre-program period is 2008–2010, and the post-program period is 2011–2013. The implementation of GECP leads to a 3.2% increase in NDVI (Table 1, Column 1). We tested the parallel-trend assumption using an event study analysis

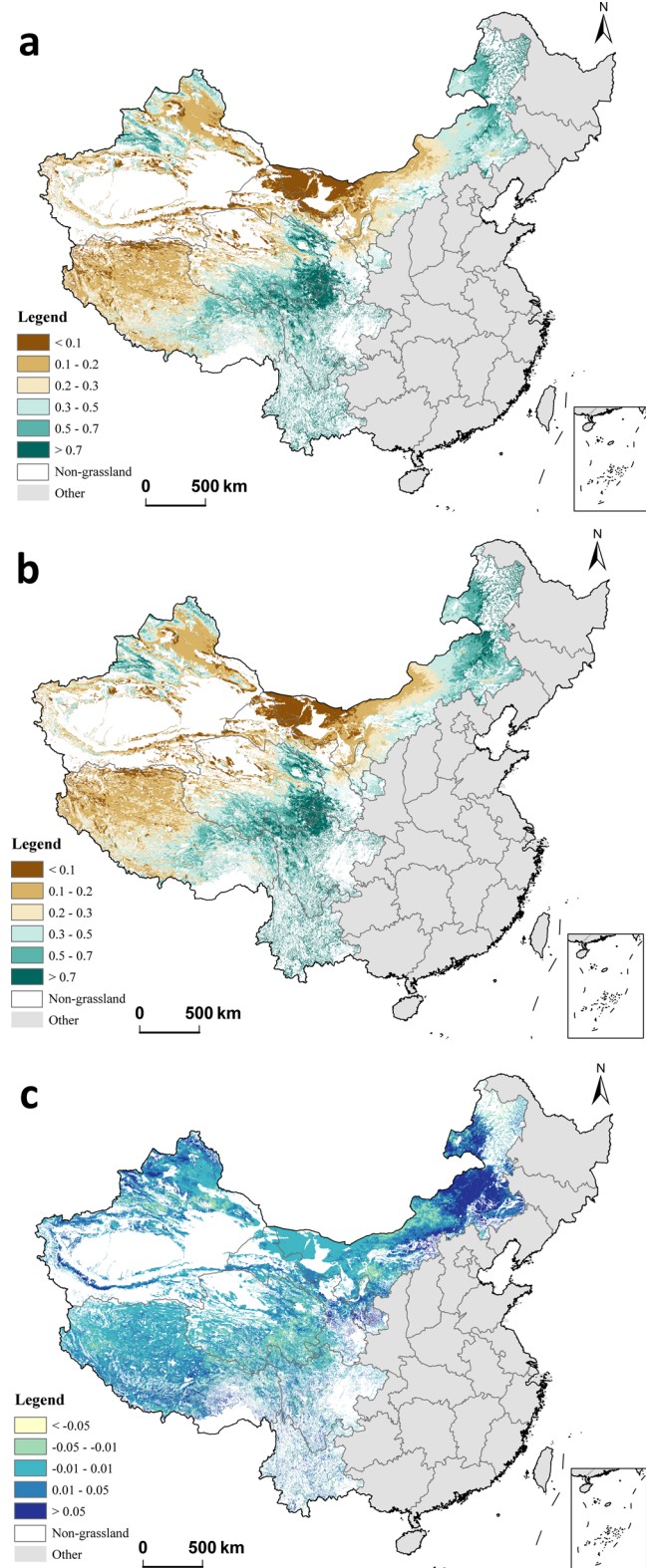

**Fig. 1 Temporal and spatial trends of NDVI. a** Average before Grassland Ecological Compensation Program (GECP) implementation (2006−2010); **b** average NDVI after GECP implementation (2011-2015); **c** difference in NDVI before and after GECP implementation. NDVI stands for the Normalized Difference Vegetation Index, which is a measure for grassland quality. It was constructed based on infrared and near-infrared channel remote sensing images and has been widely used as an indicator of vegetation coverage.

(Supplementary Fig. 1), which suggests that the assumption is valid. We also employed numerous robustness checks, such as selecting different time periods and different treatment and control groups for analysis (Supplementary Tables 2 and 3). The results remained robust when using a random-effects (RE) model (Table 1, Column 2). These checks confirmed our main finding that GECP led to positive, but small, impacts. We also found that GECP started to have positive impacts on improving grassland quality about two years after program implementation (Supplementary Table 3).

We also employed an FE model and household-level data from a survey conducted in five pastoral provinces to evaluate the impact of payment intensity (i.e., payment amount per hectare of grassland) of GECP. This model also found significant, albeit small, impacts on grassland quality. Table 1, Column 3 shows that a 1% increase in payment intensity led to an increase of 0.011% in NDVI. In other words, if payment intensity doubles, NDVI would increase by only 1.1%. Echoing county-level estimates, this is a statistically significant, yet ecologically small, impact. The pre-trend assumption test of reverse causality revealed no systematic differences in grassland quality between households with different GECP payments in the pre-program period (2008–2010) (Supplementary Table 4), indicating that our key explanatory variable (payment intensity) is exogenous to grassland quality. That is, payment intensity affects grassland quality, but does not depend on grassland quality before GECP implementation, lending further support to our findings.

**Impacts on income and income inequality among herders.** Since implementation, GECP payments have become a major income source for herder households (Fig. 2a). Taking Qinghai and Gansu as an example, annual household income per capita in 2017 was 13,136 RMB, of which 28% (3,661 RMB) came from direct GECP payments; 56% (7,411 RMB), from pastoral-sector income; and 16% (2,064 RMB), from non-pastoral-sector income. In contrast, annual household income per capita in 2010 (the year immediately prior to program implementation) was 9,100 RMB, of which nearly 90% came from the pastoral sector, while the remaining 10% came from non-pastoral sectors. This comparison suggests that annual household income per capita in 2017 was about 4,000 RMB higher than in 2010; nearly 90% of this increase (3,660 RMB) was from GECP.

We also used a household-level FE model to estimate the impact of GECP payments on total household income, net pastoral income, non-pastoral income, and non-program income. Non-program income includes net pastoral income and non-pastoral income. Panel A of Table 2 shows that the GECP program significantly raised total household income but had little effect on their pastoral income, non-pastoral income, and non-program income. The FE models estimated that a 10% increase in annual GECP payments led to an increase of 3.66% in total household income per capita (Column 1). Although the coefficients of pastoral income (Column 2), non-pastoral income (Column 3), and non-program income (Column 4) are statistically insignificant, they are all positive. This indicates that, although the GECP program has an overall impact on herder income, in general, the program did not boost any specific types of income. This may be due to the fact that there are differences in the emphasis on different types of specific sources of income in the different parts of the sample.

Although GECP increased herder income overall, it seemed to exacerbate existing income inequality. By dividing the sample into three terciles based on household income in 2010, we found that, although GECP payments represented a smaller percentage of household income for the high-income group, this group received

**Table 1 Estimated impact of the Grassland Ecological Compensation Program (GECP) on grassland quality: county and household levels.**

| Models | (1) County level | (2) | (3) Household level |
|---|---|---|---|
| | DID | RE | FE |
| $P \times T$ | 0.032*** | 0.024*** | |
| | (0.000) | (0.000) | |
| Log (payment intensity) (yuan/ha) | | | 0.011* |
| | | | (0.058) |
| Number of observations | 3,425 | 3,425 | 3110 |
| Number of counties/households | 574 | 574 | 821 |
| $R^2$ | 0.989 | 0.806 | 0.958 |

Note. The dependent variable is NDVI in log form. NDVI stands for the Normalized Difference Vegetation Index, which is a measure for grassland quality. It was constructed based on infrared and near-infrared channel remote sensing images and has been widely used as an indicator of vegetation coverage. Column (1) provides the results from the DID (difference-in-differences) approach using county-level data (Eq. (1)). The treatment group ($P = 1$) includes the counties in five North and Northwestern program provinces that were covered in GECP-I, i.e., Xinjiang, Qinghai, Gansu, Ningxia, and Inner Mongolia. The control group ($P = 0$) includes the counties in five North and Northeastern provinces that were not covered by GECP-I (i.e., the first five-year period of GECP) until 2016, i.e., Shanxi, Hebei, Liaoning, Jilin, and Heilongjiang. The pre-program period ($T = 0$) is 2008–2010. The post-program period ($T = 1$) is 2011–2013. Column (2) presents the results from the county-level random-effect (RE) model. In Columns (1) and (2), year and province fixed effects are controlled for. Climate controls include monthly rainfall, temperature, and the Palmer Drought Severity Index (PDSI) for May–October. Socioeconomic control includes per-capita county fiscal income. Robust standard errors are clustered at the county level. Column (3) reports the results from the household-level fixed-effect (FE) model (Eq. (3)). Both household and year-fixed effects are controlled for. Household-, village-, and township-level time-variant variables also are controlled for. Household-level controls include the quantity of labor used in raising livestock, operated farm size, share of the joint operated area, the total number of different plots, a dummy variable for grassland harvesting, and a dummy variable for planting crop/fodder. Village-level controls include an indicator of whether a village has local grassroots measures in place to limit grazing intensity, an indicator of whether a village has a formal government-run monitoring system, and climate variables (cumulative rainfall and mean temperature from May to October in each year). Township-level controls include farm-gate livestock prices, hay prices, wages for non-pastoral employment, and grassland rental prices. Standard errors are clustered at the village-by-year level. A two-sided $t$ test is performed for each coefficient. The exact $p$-values are in parentheses. *$p < 0.10$, ***$p < 0.01$.

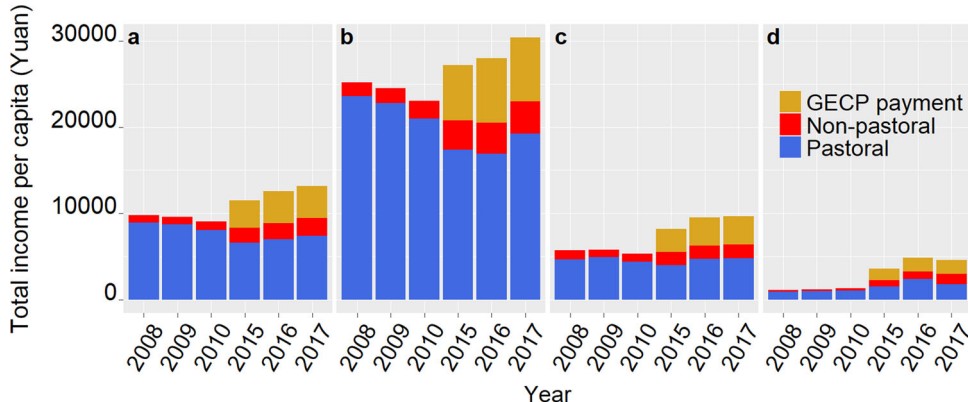

**Fig. 2 Decomposition of household income.** The samples in Qinghai and Gansu are divided into three terciles, based on the 2010 household income per capita. **a** Total income per capita for the whole sample in Qinghai and Gansu; **b** total income per capita for high-income group; **c** total income per capita for middle-income group; **d** total income per capita for low-income group.

significantly more payments from GECP (Fig. 2b, c, d). Annual household income per capita for the high-income group increased by about 7,300 RMB between 2010 and 2017. Almost all of the increase can be attributed to GECP payments. In contrast, household income per capita for the low-income group increased by 3,250 RMB between 2010 and 2017, for which only about 50% (approximately 1,600 RMB) came from GECP payments. Importantly, GECP payments per capita received by the high-income group were more than four times those paid to the low-income group (7,440 RMB versus 1,590 RMB).

Our FE estimates confirm this trend of the widening of income inequality. A 1% increase in annual household GECP payments led to an increase of 0.45% in household income for the low-income group; 0.37% for the medium-income group; and 0.34% for the high-income group (Table 3, Column 1). These estimates imply that doubling GECP payments increased household income per capita by 407 RMB for the low-income group, 1,470 RMB for the medium-income group, and 6,955 RMB for the high-income group. Even though GECP payments

represented a smaller portion of total income for the high-income group, in absolute terms, high-income households received more money from the program.

An examination of the subcategories of income reveals that, although GECP payments had virtually no effect on any income group (Table 3, Column 2), non-pastoral income increased for the high-income group (Table 3, Column 3) but only by a small amount. A 1% increase in total payments led to a non-pastoral income increase of 0.09% for the high-income group. In other words, doubling the total payments would have resulted in an annual rise in non-pastoral income of 180 RMB per capita for the high-income group. Given that the average herder income before the program was already 9,100 RMB per capita, these are not substantially large increases.

**Mechanisms of the effect on grassland quality.** Understanding the mechanisms of GECP may allow policymakers to improve the program design. We, therefore, explored potential channels through which GECP affected grassland quality. We examined

**Table 2 Estimated impacts of the Grassland Ecological Compensation Program (GECP) on herder income: household level, FE model.**

| Dependent variable | (1) Household income per capita | (2) Net pastoral income per capita | (3) Non-pastoral income per capita | (4) Non-GECP income per capita |
|---|---|---|---|---|
| **Panel A. Overall impacts on income** | | | | |
| Annual GECP payment (yuan) | 0.366*** (0.004) | 0.132 (0.369) | 0.048 (0.366) | 0.144 (0.332) |
| Control variables | Yes | Yes | Yes | Yes |
| Household fixed effects | Yes | Yes | Yes | Yes |
| Year fixed effects | Yes | Yes | Yes | Yes |
| No. of observations | 3,474 | 3,474 | 3,474 | 3,474 |
| $R^2$ | 0.731 | 0.749 | 0.863 | 0.744 |
| **Panel B. Heterogeneous impacts by grassland types** | | | | |
| Grassland | 0.405* (0.051) | 0.221 (0.347) | 0.025 (0.746) | 0.179 (0.445) |
| Meadow | 0.232* (0.057) | −0.243 (0.131) | 0.122** (0.030) | −0.043 (0.777) |
| Desert | 0.274 (0.414) | 0.309 (0.334) | 0.014 (0.873) | 0.300 (0.345) |
| **Panel C. Heterogeneous impacts by socioeconomic variables** | | | | |
| Education of the labors (years) | −0.004 (0.183) | −0.002 (0.473) | 0.020*** (0.000) | −0.001 (0.852) |
| Distance to the closest township-level road (km) | −0.0002 (0.654) | −0.0000 (0.978) | −0.001** (0.011) | −0.0001 (0.742) |
| Grassland area per capita (hundred ha in log) | 0.009 (0.438) | 0.016 (0.344) | −0.035*** (0.000) | 0.010 (0.467) |

*Note.* Panel A provides the estimated overall impacts on income from the fixed-effect (FE) model using household-level data (Eq. (3)). All outcome variables and the key explanatory variable, annual GECP payment, are transformed using an inverse hyperbolic sine transformation to avoid taking logarithm of zero, following $\ln(y+(y^2+1)^{1/2})$. Household income includes net pastoral income, non-pastoral income, and GECP payment. Both household and year-fixed effects are controlled for. Household-, village-, and township-level time-variant variables are also controlled for. Household-level controls include the quantity of labor used in raising livestock, operated farm size, share of the joint operated area, the total number of different plots, a dummy variable for grassland harvesting, and a dummy variable for planting crop/fodder. Village-level controls include an indicator of whether a village has local grassroots measures in place to limit grazing intensity, an indicator whether a village has a formal government-run monitoring system, and climate variables (the cumulative rainfall and the mean temperature from May to October in each year). Township-level controls include farm-gate livestock prices, hay prices, wages for non-pastoral employment, and grassland rental prices. Standard errors are clustered at the village-by-year level. The results are consistent using an inverse hyperbolic sine transformation and a log transformation of the dependent variable in each column. The log transformation results are available upon request. Panel B provides the heterogeneous impacts on income by grassland type. We use the grassland type with the largest area in a county as its major grassland type. Each set of coefficients and corresponding p-values are from one single regression model. All model specifications are the same as in Panel A. Panel C provides the heterogeneous impacts on income by different socioeconomic variables. A two-sided *t* test is performed for each coefficient. The exact p-values are in parentheses. *p < 0.10, **p < 0.05, ***p < 0.01.

**Table 3 Estimated impacts of the Grassland Ecological Compensation Program (GECP) payment on income equity: household level, fixed-effect (FE) model.**

| Dependent variable | (1) Household income per capita | (2) Net pastoral income per capita | (3) Non-pastoral income per capita |
|---|---|---|---|
| Low-income group | 0.452*** (0.001) | 0.103 (0.493) | −0.053 (0.332) |
| Middle-income group | 0.365*** (0.007) | 0.129 (0.395) | 0.026 (0.627) |
| High-income group | 0.337*** (0.010) | 0.142 (0.326) | 0.086* (0.063) |
| Number of observations | 3,469 | 3,469 | 3,469 |
| $R^2$ | 0.732 | 0.749 | 0.865 |

*Note.* This table provides the results from the FE model using household-level data (Eq. (6)). All dependent variables and the key independent variable, annual GECP payment, are taken as an inverse hyperbolic sine transformation to define zero numbers, following $\ln(y+(y^2+1)^{1/2})$. Household income includes net pastoral income, non-pastoral income, and GECP payment. Both household and year-fixed effects are controlled for. Household-, village-, and township-level time-variant variables also are controlled for. Household-level controls include the quantity of labor used in raising livestock, operated farm size, share of the joint operated area, the total number of different plots, a dummy variable for grassland harvesting, and a dummy variable for planting crop/fodder. Village-level controls include an indicator of whether a village has local grassroots measures in place to limit grazing intensity, an indicator of whether a village has a formal government-run monitoring system, and climate variables (cumulative rainfall and mean temperature from May to October in each year). Township-level controls include farm-gate livestock prices, hay prices, wages for non-pastoral employment, and grassland rental prices. Standard errors are clustered at the village-by-year level. A two-sided *t* test is performed for each coefficient. The exact p-values are in parentheses.
*p < 0.10, ***p < 0.01.

**Table 4 Estimated impacts of the Grassland Ecological Compensation Program (GECP) on livestock production: county level.**

| Dependent variable | (1) Log (year-end cattle inventory) | (2) Log (year-end sheep inventory) |
|---|---|---|
| P × T | −0.027 (0.602) | −0.121** (0.019) |
| Number of observations | 3,992 | 4,000 |
| $R^2$ | 0.810 | 0.800 |

*Note.* This table provides the results from the difference-in-differences (DID) approach for the estimated impacts of the GECP on livestock production using county-level data (Eq. (1)). NDVI stands for the Normalized Difference Vegetation Index, which is a measure for grassland quality. It was constructed based on infrared and near-infrared channel remote sensing images and has been widely used as an indicator of vegetation coverage. The treatment group (P = 1) includes the counties in five North and Northwestern program provinces that were covered in GECP-I (i.e., the first five-year period of GECP), i.e., Xinjiang, Qinghai, Gansu, Ningxia, and Inner Mongolia. The control group (P = 0) includes the counties in five North and Northeastern provinces that were not covered by GECP-I (i.e., the first five-year period of GECP) until 2016, i.e., Shanxi, Hebei, Liaoning, Jilin, and Heilongjiang. The pre-program period (T = 0) is 2008–2010. The post-program period (T = 1) is 2011–2013. Year and province fixed effects are controlled for. Climate controls include monthly rainfall, temperature, and the Palmer Drought Severity Index (PDSI) for May to October in each year. Socioeconomic control includes per-capita county fiscal income. Robust standard errors are clustered at the county level. A two-sided *t* test is performed for each coefficient. The exact p-values are in parentheses.
**p < 0.05.

changes in livestock scale, supplementary feeding, and operated grassland area. We found that herders reduced livestock inventory as a response to the GECP. However, further increasing payment standards without augmenting other aspects of implementation would not reduce livestock production, but may encourage herders to enlarge farm size by renting in grassland and increase supplementary feedings as responses.

The analysis at the county level shows that herders reduced sheep inventory but did not reduce cattle inventory as a response to the GECP program. Applying the DID approach to county-level livestock data, we found that GECP reduced year-end sheep inventory by 12.1%, but the effect on cattle inventory was statistically insignificant (Table 4). Estimates derived from household livestock data show that an increase in the per-hectare GECP payment had no significant effect on cattle inventories (Table 5, Columns 1–3). This indicates that further

**Table 5 Estimated impacts of the Grassland Ecological Compensation Program (GECP) on herder behavior: household level.**

| Dependent variable | (1) Livestock inventory | (2) Cattle inventory | (3) Sheep inventory | (4) Supplementary feeding | (5) Grassland rent in |
|---|---|---|---|---|---|
| $Log$ (payment intensity (yuan/ha)) | −0.003 (0.896) | −0.032 (0.359) | 0.021 (0.520) | 0.075** (0.024) | 0.012** (0.020) |
| Number of observations | 3,429 | 3,429 | 3,429 | 3,473 | 3,474 |
| $R^2$ | 0.723 | 0.898 | 0.938 | 0.826 | 0.892 |

*Note.* This table reports the results from the household-level fixed-effect (FE) model (Eq. (3)). All dependent variables are transformed using a hyperbolic sine transformation to avoid taking logarithm of zero, following $\ln(y+(y^2+1)^{1/2})$, with the exception of the variable indicating grassland rent in or not. The payment intensity is log-transformed. Both household and year-fixed effects are controlled for. Household-, village-, and township-level time-variant variables are also controlled for. Household-level controls include the quantity of labor used in raising livestock, operated farm size, share of the joint operated area, the total number of different plots, a dummy variable for grassland harvesting, and a dummy variable for planting crop/fodder. Village-level controls include a variable that indicates whether a village has local grassroots measures in place to limit grazing intensity, a variable that indicates whether a village has a formal government-run monitoring system, and climate variables (cumulative rainfall and mean temperature from May to October in each year). Township-level controls include farm-gate livestock prices, hay prices, wages for non-pastoral employment, and grassland rental prices. Standard errors are clustered at the village-by-year level. A two-sided $t$ test is performed for each coefficient. The exact $p$-values are in parentheses. **$p < 0.05$.

increasing the level of payment does not appear to reduce livestock inventories, given the current approach to implementing the program (that is, without augmenting the current program with other measures).

Table 5, Column 4 shows that GECP payments were significantly correlated with supplementary feeding at the household level. The results indicate that a 10% increase in payment per hectare leads to a 0.75% increase in supplementary feeding. This suggests that herders increased supplementary feeding as a response to GECP payments, but did so at a low level. Unfortunately, we cannot confirm this at the county level, as county-level data on supplementary feeding do not currently exist.

When considering the household level, the analysis also demonstrates that herders enlarged their operated grassland area in response to the GECP program. Table 5, Column 5 shows that a 10% increase in payment per hectare leads to a 0.12% increase in the likelihood of renting in grassland. The effect, which is statistically significant, suggests that there is a grassland rental market between herders, although the small magnitude of the estimate suggests that this market may be incomplete or underdeveloped. Unfortunately, data limitations hinder our ability to further examine the impacts of the program on the total grassland area; similar to our analysis of supplementary feeding, we lack county-level data to compare with our household-level results.

**Heterogeneity analysis**. The heterogeneity analysis can, in some cases, be used by policymakers to propose supporting measures that may improve the GECP program and allows researchers to identify key factors that may affect the effectiveness of GECP. As such, we explored several dimensions in which GECP may have heterogeneous impacts. At the county level, we examined the effect of market accessibility, proxied by rural road intensity and grassland quality prior to GECP implementation. We also examined the spatially heterogeneous impacts, captured by the effects on counties classified as having primarily different types of grassland. At the household level, we examined the roles of the per-capita grassland area, non-pastoral wage level, local grassroots measures, and formal monitoring system in driving the heterogeneous ecological impacts. We also examined heterogeneous impacts on herder income across different grassland types reflecting spatial heterogeneity, and some socio-economic variables including household laborers' years of education, distance to township-level roads, and per-capita grassland area.

At the county level, we first analyzed how road density affects the effectiveness of GECP. Specifically, more intense road networks imply greater market access, allowing herders to sell livestock, purchase supplementary feed, and work in non-pastoral sectors. In our analysis, road density is measured by the length of road per square kilometer. To conduct the analysis, we split the sample into three groups by road density: low-density group, with only one 40-m road per kilometer square km$^2$, and medium- and high-density groups with road densities of 220 m/km$^2$ and 380 m/km$^2$, respectively. Applying the DID approach to the county-level data, we found that GECP has a larger NDVI-improving impact in counties with greater road intensity, with 5.3% for the high-density group, 2.5% for the medium-density group, and 1.7% for the low-density group (Table 6, Panel A).

Second, the county-level heterogeneous analysis examined the role of initial grassland quality. We divided the sample into three groups according to the grassland quality (i.e., low, medium, and high level) in 2008, a year before GECP implementation. We found that counties with medium-level grassland quality prior to GECP implementation experienced the largest ecological improvements. NDVI increased by 3.5% in counties with low NDVI, 5.5% for medium-NDVI counties, and 2.6% for high-NDVI counties (Table 6, Panel B). This suggests that GECP is more effective in protecting and improving moderately degraded grasslands rather than endangered grasslands. Further analysis is needed to determine the reasons for this heterogeneity, as areas with poor grassland quality may require either specific improvements in the GECP program or new policies to be developed.

Third, we explored potential spatial heterogeneity as related to the impact of the GECP program. We used grassland type as an indicator for spatial heterogeneity, under the assumption that different types of grassland reflect different natural resource endowments and affect the nature of each region's economic activities. Following Ma and Xu[22], we adopted the classification of five grassland types (i.e., meadow, grassland, desert, shrubland, and herbosa), which are all covered in our county-level data.

The results (Table 6, Panel C) suggest that, although there is some spatial heterogeneity based on the nature of the resource endowments of counties, the effects are mainly consistent across space: GECP has had a positive effect on grassland quality in most counties (classified as having different types of grassland) in the program area. More specifically, the program raised grassland quality in all counties (including counties classified as having primarily grassland, desert, shrubland, and herbosa), with the exception of counties that are classified as having primarily meadows. The coefficients of the GECP program variable vary across the four areas in which statistically significant impacts were found (from 0.018 for shrubland to 0.055 for grassland). The coefficient (of the GECP program variable) for counties with meadows also is positive, although the point estimate is small (and, as stated, statistically insignificant).

Complementing the results of the county-level analysis, the household-level heterogeneous analysis found that GECP had a larger positive impact on grassland quality for large farms and those with local grassroots measures and/or formal government-run monitoring systems to limit grazing intensity. First, we

**Table 6 Estimated heterogeneous impacts of the Grassland Ecological Compensation Program (GECP) on grassland quality: county level.**

| | (1) | | (2) | | (3) |
|---|---|---|---|---|---|
| | **Low** | | **Medium** | | **High** |
| *Panel A by rural road intensity in 2008:* | | | | | |
| $P \times T$ | 0.017 | | 0.025*** | | 0.053*** |
| | (0.157) | | (0.011) | | (0.000) |
| Number of observations | 1,122 | | 1,144 | | 1,149 |
| $R^2$ | 0.994 | | 0.988 | | 0.977 |
| | Low | | Medium | | High |
| *Panel B by NDVI in 2008:* | | | | | |
| $P \times T$ | 0.035*** | | 0.055*** | | 0.026*** |
| | (0.002) | | (0.000) | | (0.001) |
| Number of observations | 1,128 | | 1,150 | | 1,086 |
| $R^2$ | 0.991 | | 0.966 | | 0.948 |
| | Meadow | Grassland | Desert | Shrub land | Herbosa |
| *Panel C by grassland type:* | | | | | |
| $P \times T$ | 0.001 | 0.055*** | 0.030*** | 0.018** | 0.038*** |
| | (0.850) | (0.000) | (0.000) | (0.046) | (0.000) |
| Number of observations | 2,038 | 2,380 | 2,231 | 1,894 | 1,714 |
| $R^2$ | 0.969 | 0.979 | 0.995 | 0.952 | 0.953 |

*Note.* The dependent variable is NDVI in log form. NDVI stands for the Normalized Difference Vegetation Index, which is a measure for grassland quality. It was constructed based on infrared and near-infrared channel remote sensing images and has been widely used as an indicator of vegetation coverage. In Panels A and B, we present the results when we first divide the control and treatment groups into three terciles (i.e., low, medium, and high), based on each indicator in a base year (i.e., 2008). In Panel C, all of the control and treatment counties are grouped into five grassland types. We then pair the subgroups in the control and treatment groups and run the model indicated by Eq. (1). All other model specifications are the same as in Column (1) of Table 1. A two-sided *t* test is conducted for each coefficient. The exact *p*-values are in parentheses.
**p < 0.05, ***p < 0.01.

**Table 7 Estimated heterogeneous impacts of the Grassland Ecological Compensation Program (GECP) on grassland quality: household level.**

| Heterogeneous variables | (1) By per-capita grassland area | (2) By local grassroots measures (yes = 1, no = 0) | (3) By formal monitoring system (yes = 1, no = 0) | (4) By non-pastoral wage |
|---|---|---|---|---|
| *Log* (payment intensity (yuan/ ha) (c1) | 0.037*** | 0.008 | 0.008 | 0.013* |
| | (0.000) | (0.215) | (0.158) | (0.077) |
| Interaction term (c2) | 0.016** | 0.009** | 0.006*** | 0.0006 |
| | (0.000) | (0.047) | (0.010) | (0.762) |
| $H_0$: c1 + c2 = 0 | - | 0.017** | 0.014** | - |
| Number of observations | 3,093 | 3,110 | 3,110 | 3,110 |
| $R^2$ | 0.959 | 0.959 | 0.960 | 0.958 |

*Note.* This table provides the results for heterogeneous effects of GECP on grassland quality using household-level data (Eq. (5)). The dependent variable is NDVI in log form. NDVI stands for the Normalized Difference Vegetation Index, which is a measure for grassland quality. It was constructed based on infrared and near-infrared channel remote sensing images and has been widely used as an indicator of vegetation coverage. Both household and year-fixed effects are controlled for. Household-, village-, and township-level time-variant variables also are controlled for. Household-level controls include the quantity of labor used in raising livestock, operated farm size, share of the joint operated area, the total number of different plots, a dummy variable for grassland harvesting, and a dummy variable for planting crop/fodder. Village-level controls include an indicator of whether a village has local grassroots measures in place to limit grazing intensity, an indicator of whether a village has a formal government-run monitoring system and climate variables (cumulative rainfall and mean temperature from May to October in each year). Township-level controls include farm-gate livestock prices, hay prices, wages for non-pastoral employment, and grassland rental prices. Standard errors are clustered at the village-by-year level. A two-sided *t* test is performed for each coefficient. The exact *p*-values are in parentheses. *p < 0.10, **p < 0.05, ***p < 0.01.

analyzed the role of farm size. The results suggest that farm size was positively related to the improvement of grassland quality. Specifically, as farm size (as indicated by grassland area per capita) increased by 1%, the GECP impact increased by 0.016 percentage points (Table 7, Column 1). Put differently, doubling farm size would have increased NDVI by 1.64 percentage points.

Second, GECP had a larger NDVI-improving impact in the villages with some type of accountability in place to limit grazing intensity. This accountability can come from local grassroots measures (Table 7, Column 2) or formal government-run monitoring systems (Table 7, Column 3). Local grassroots measures are referred to as measures by which local rules or norms, usually enforced informally by village elders, limit the total number of sheep units that can be grazed in an area. When local grassroots measures are in place, a 1% increase in GECP payments led to a 0.9% increase in NDVI. In contrast, GECP did not significantly increase grassland quality in areas without local grassroots measures. We see a similar trend with formal government monitoring systems. Areas with GECP that did have formal government monitoring experienced a 0.6% increase in NDVI, whereas GECP did not have a significant impact in areas without formal government monitoring. These contrasts suggest

that having some system of accountability or monitoring, regardless of whether it is a formal or informal system, may be essential to program success.

Finally, we included non-pastoral wages as a proxy for the opportunity costs of pastoral labor. High non-pastoral wages may attract more herders to work outside the pastoral sector. We found that, surprisingly, non-pastoral wages do not significantly affect the GECP impact on grassland quality (Table 7, Column 4). One possible reason is that low levels of education and human capital may prevent herders from obtaining high-paying non-pastoral jobs. Our field experiences show that nearly half of the herders did not have any formal education, and their mean education level was only 3.5 years. The observed low levels of human capital provide some explanation for why high non-pastoral wages (and, thus, high opportunity costs for pastoral labor) seem to have no effect on the GECP impact in terms of improving grassland quality.

In addition to the heterogeneous impacts on grassland quality, we also examined the heterogeneous impacts on herder income by different grassland types that reflect spatial heterogeneity (Panel B of Table 2) and certain socioeconomic variables (Panel C of Table 2). We found that the GECP program has different impacts on annual household income per capita in areas with different grassland types and that the impacts on non-pastoral income per capita vary across groups with different socioeconomic characteristics.

First, GECP has a larger impact with regard to improving household income per capita in grassland and meadow, whereas the impact in the desert is insignificant (Column 1 of Panel B). Household income per capita has been improved by 40.5% in grassland and 23.2% in meadow regions. This is due to the regulatory standard implemented by the program. As noted above, regions with high baseline grassland quality, such as grassland and meadow regions, are assigned a higher payment standard to compensate for higher levels of lost income from the livestock sector. Regions with low grassland productivity, such as desert areas, in contrast, are assigned a lower payment standard.

Second, the impacts on non-pastoral income are heterogeneous across groups with different socioeconomic characteristics (Column 3 of Panel C), whereas the impacts on household income, pastoral income, and non-program income are more homogeneous (Columns 1, 2, and 4 of Panel C). The GECP program has a larger impact on non-pastoral income for herders with higher levels of education, those who live closer to the township-level road, and those with smaller grassland area per capita. This indicates that herders with more education are probably more competitive in non-pastoral labor markets. Herders who live closer to roads are more likely to obtain a non-pastoral job, as the travel cost is lower and job market information may be more accessible. Herders with fewer grassland areas per capita are more likely to switch to a non-pastoral job.

## Discussion

This study sought to comprehensively evaluate the GECP program based on its stated objectives. To this end, we combined remote sensing and household survey data to assess GECP impacts on three outcome metrics: grassland quality, herder income, and income distribution. In addition, to gain a better understanding of GECP impacts, and, thus, be able to suggest improvements to the program, we also explored potential mechanisms and heterogeneous impacts of GECP. Although we discuss them more in this section, in brief, our results show that GECP only slightly improved grassland quality and significantly increased herder income, yet

exacerbated income inequality among herders. This suggests that improvements should be made to enhance the program's effect on both grassland quality and herder livelihood. Although the heterogeneity analysis showed that the conclusions may differ to some extent across space and groups with different socioeconomic characteristics, these conclusions still hold in general as the implementation of the program is quite uniform and homogeneous. The heterogeneity results highlight the importance of suiting flexible and bottom-up implementations to local circumstances.

Our overall results suggest that GECP implementation only slightly improved grassland quality, which is at odds with the findings of Liu et al.[21], one of the only two other studies that examined GECP impacts. Liu et al. estimated that GECP increased grassland quality by about 30%, whereas our results show that GECP led to only a roughly 5% increase. This divergence in results likely stems from differences in methodology. In particular, Liu et al. lack proper control groups, perhaps leading them to overestimate the impact of GECP. As our study overcomes this limitation, it represents a methodological improvement over that of Liu et al.

Our results show that GECP has been able to improve grassland quality by reducing livestock production. In fact, these findings are consistent with those found in Hu et al.[20]. In their survey in Inner Mongolia, the research team found that GECP payments prompted farmers to reduce sheep herds but not cattle herds. However, we also found that increasing the level of payment does not appear to reduce livestock production, but instead encourages herders to enlarge farm size by renting in grassland and increase supplementary feedings as responses. As herders rented out their grassland, their labor was (at least partially) released from the pastoral sector, allowing them to take on non-pastoral jobs. Therefore, it is through the non-pastoral jobs that the pressure on the grassland was reduced, as fewer households were relying on the grassland to make a living. Herders also used more supplementary feeding to release livestock pressure on grassland, but did so at a low level, which may reflect the incompleteness of hay markets or constraints on cash liquidity.

This study also points to two potential channels to make GECP more effective at improving grassland quality. First, our results indicate that investing in rural roads may help to enhance the impacts of GECP on grassland quality. The literature shows that higher density of rural roads can generate many benefits, including increases in market participation[23], non-farm income[24], and economic development[25]. In this case, increased market accessibility may facilitate herders to sell livestock, purchase supplementary feed, and work in non-pastoral sectors, thereby reducing everyday reliance on grasslands and allowing the grasslands time to recover. Better transportation also may boost the development of tourism and bring in more income to herders[26].

Second, both formal government-run monitoring systems and local grassroots measures enhance the effects of GECP in improving grassland quality. Ineffective monitoring and sanctions may cause some ecosystem service suppliers to become non-compliant. Unfortunately, current GECP monitoring and sanctioning systems are labor-intensive and far from effective. Formal government-run monitoring systems are expensive but have similar impacts as do local measures, suggesting that informal monitoring may be more cost-effective. Further, informal systems have inefficiencies due to (dis)economy of scale: One monitoring person, usually an elected villager, is responsible for monitoring about 2,700 hectares of grassland. Such a workload is too large to accomplish efficiently. Because external governance faces high transaction costs (e.g., costs of supervision), empowering or

expanding informal systems (e.g., self-governance) may be a cost-effective way to enhance monitoring.

GECP efficiency could be further improved by better applying PES theory to the program. Wunder[2] provides the five criteria of a successful PES program:

(1) A voluntary transaction in which (2) a well-defined environmental service (or a land-use likely to secure that service) (3) is "bought" by a (minimum of one) buyer (4) from a (minimum of one) provider (5) if and only if the provider continuously secures the provision of the service (conditionality) (p. 50).

Regrettably, the uniform application and universal enrollment of the GECP program limits our ability to evaluate several key factors implied by PES theory, specifically, voluntary enrollment and conditional performance. GECP payments are not related to grassland quality, which violates the criteria of conditionality. A lack of conditionality weakens herder incentives to change behavior, as herders can still gain GECP transfers without changing behavior. This could explain not only the low NDVI gains from the program overall but also why the program is effective only when some sort of monitoring is in place; that is, herders seem to change behavior only when there is some type of monitoring system (formal or informal) in place to hold them accountable. Voluntary participation is also important, as it makes monitoring much more efficient and cost-effective. As stated above, the current monitoring systems are extremely inefficient and inadequate. Voluntary participation, as required by PES theory, would allow certain herders to opt-out, reducing the overall burden of monitoring. Thus, a better application of PES theory to GECP would enhance program effectiveness by cost-effectively increasing monitoring efficiency.

Our results also indicate that GECP increases herder income while, notably, also exacerbating income inequality among herders. This is because GECP payments are designed to compensate for lost income per unit of land. Because richer households usually have more land (and, therefore, lose more income potential from not grazing their lands), they receive larger GECP payments even though the payment per land unit is the same within a county. More external money helps herders reduce their living pressure on the grassland ecosystem; therefore, grassland quality is better improved in farms with larger operated grassland areas than in those with small ones. This indicates a tradeoff between efficiency and equity[1,11]: allocating limited funding to larger farms can gain ecological efficiency but will exacerbate local income inequality. This widened income inequality, in turn, hurts society and, thus, may even hinder herder livelihood.

The results of this study have three policy implications. First, policymakers should pay more attention to policy externalities, such as enlarged income inequality, as they may undercut potential program gains. Tradeoffs between ecological and income gains and income equality need to be balanced. Second, GECP policy design should follow PES theory more closely, as this would enhance monitoring and program efficiency. Third, to make GECP more effective at achieving its goals, it is necessary to provide complementary supporting measures. Barriers for herders to work in non-pastoral sectors could be removed through, for example, improving herders' human capital and building more roads to facilitate transportation and market access; this may relieve pressure on the grassland, as fewer people's livelihood will rely on it. In addition, employing local grassroots measures to limit grazing intensity may represent a more efficient way to increase monitoring than do formal government-run monitoring systems.

This study contributes to the literature in two ways. First, it provides more evidence of the impact of GECP, which is, by far, the largest PES program focusing on grasslands in the world[20,21].

To the best of our knowledge, this is one of only three studies that evaluate such a large grassland protection program. We believe that our study also is more comprehensive than either previous study. We not only investigate both of the program's explicit policy objectives but also examine income equity, which is an important socioeconomic measure related to societal fairness. This study also examines the working mechanisms of the program and its heterogeneous impacts, which deepens our understanding of the program's effects. Second, this study fills existing gaps in the PES literature. Few PES projects have focused on conserving grasslands, and even fewer studies have assessed these projects. These insights of this study could be especially helpful in informing conservation policy in other developing countries with similarly degraded grasslands.

Although this study provides important contributions to the literature, it has certain limitations. We acknowledge that our income data may suffer from measurement errors, as our income variable is constructed based on recall data. Fortunately, not all of our variables rely on recall data: variables that measure grassland quality are based on remote sensing images, and GECP payments are based on herder bank records, neither of which relies on recall data, and are, therefore, less likely to suffer from measurement errors. In addition, we estimated both FE and RE models as robustness checks against attenuation bias due to measurement errors in income data (as RE models would suffer less from attenuation bias, if it exists). The results from RE models are presented in Supplementary Table 9 and are mostly consistent with the results from our FE models.

## Methods

The evaluation of GECP impacts is based on two sets of data: county level and household level. Although both are panel data sets, given their different structures, different model specifications were adopted for data analysis. Below, we first introduce the two data sets and then discusses empirical models for analysis at the two levels. To evaluate the impacts on grassland quality at the county level, we use a DID approach. To evaluate both grassland quality and herder income at the household level, we use FE models.

### Data

*County-level data*. To evaluate the impact of GECP on grassland quality, we compiled data from different sources to create a county-level data set. The data set covers 630 GECP program counties in eight provinces (Xinjiang, Tibet, Qinghai, Gansu, Ningxia, Inner Mongolia, Sichuan, and Yunnan) and 386 non-program counties in five provinces (Shanxi, Hebei, Liaoning, Jilin, and Heilongjiang) during the pre-program (2005–2010) and post-program periods (2011–2015) ((Supplementary Fig. 2). This compiled data set contains information on grassland quality, climate conditions, and aggregated socioeconomic characteristics for all 1,016 counties.

Grassland quality is measured by NDVI, which was constructed based on infrared and near-infrared channel remote sensing images and has been widely used as an indicator of vegetation coverage[27–29]. Because grassland ecosystems have a relatively simple ecological structure, the use of these images is a viable method to study grassland vegetation dynamics. Monthly NDVI data at a spatial resolution of $1 \times 1$ km$^2$ were acquired from a MOD13A3 product from NASA earth data for the period of 2000–2015. More detailed information about the data set can be found in Didan[30]. These spatial data, together with county boundary data, were used to elicit NDVI for each county. The maximum of monthly NDVIs in a year were used to obtain the yearly index. The MODIS Reprojection Tool (MRT) was employed to transform and register monthly MOD13A3 data to Albers map projection and WGS84 datum. ArcMap 10.2 was then used to obtain yearly county-level NDVI data.

Monthly climate data, which include measures such as temperature, precipitation, and drought conditions, are used as control variables in the analyses. Daily meteorological data at the county level from 2005 to 2015 were obtained from the National Meteorological Information Center of China. A widely used spatial interpolation method proposed by Thornton et al.[31] was used to impute data for those counties without national stations. A cross-validation analysis was then performed to validate the accuracy of the imputations[32]. Monthly temperature and precipitation were calculated by averaging daily values over a month. Drought is measured by the Palmer Drought Severity Index (PDSI), the most widely used index of meteorological drought, originally developed by Palmer[33] and updated by Wells et al.[34]. Daily PDSI in each county was calculated by aerologists from the Chinese Academy of Sciences and then aggregated into monthly PDSI.

The county-level socioeconomic data came from two sources. The first data source uses information from the China Database on Country-level Agricultural and Rural Indicators. This database was compiled by the Ministry of Agriculture and Rural Affairs of China and is managed by the Institute of Agricultural Information of the Chinese Academy of Agricultural Sciences. The original data were collected by each county's statistical station and then reported to upper-level statistical bureaus. We use these data to measure year-end cattle and sheep inventories, per-capita grain production, and per-capita county fiscal income. Year-end cattle and sheep inventories are used as outcome variables in the analysis of the mechanism of GECP's impact on grassland quality at the county level. Per-capita county fiscal income is used as a socioeconomic control variable. Per-capita county fiscal income was calculated by dividing county fiscal income by total population in a county.

Our second county-level data source is the Data Center for Resources and Environmental Sciences of Chinese Academy of Sciences, which reports data on basic road networks. We use the basic road network data to calculate rural road intensity by dividing the length of roads in kilometers by the area of a county (measured in kilometers squared). The summary statistics of these county-level climate and socioeconomic variables in the baseline year (2010) are reported in Supplementary Tables 5–7.

*Household-level data.* The research team conducted a household survey in five provinces in the pastoral areas of China from 2017 to 2019, i.e., Qinghai and Gansu in 2017, Inner Mongolia and Xinjiang in 2018, and Tibet in 2019. We revisited the respondents in Inner Mongolia, Xinjiang, and Tibet in 2020 either by face-to-face interview in the field or phone call. As the top five pastoral provinces, their grassland areas account for about 70% of China's total grassland area[35].

The household-level panel data include farm-level grassland quality measures (also measured by NDVI) and household-level socioeconomic variables for both the pre- (2008–2010) and post-program (2015–2017) periods. Based on the results of the survey, we measured GPS coordinates of operated grasslands during the survey and matched them with yearly NDVI data to create a farm-level grassland quality panel data set. The household-level socioeconomic measures were created based on answers provided to the enumerators by the household head.

To identify and choose the sample from the provinces, we adopted a stratified random sampling strategy. We selected four counties in Gansu, five counties in Inner Mongolia, and six counties in Qinghai, Xinjiang, and Tibet, respectively. We specified three major grassland types in each province, divided all counties in each grassland type into two groups according to their annual income per capita, and randomly selected one county from each quantile. We ended up with six sample counties except Gansu and Inner Mongolia. Gansu has a small area and only has an alpine meadow as its major grassland type. We, therefore, divided all the counties in Gansu into four quantiles according to their annual income per capita and randomly selected one county from each quantile. We lost one county in Inner Mongolia as most herders in this county have quit grazing and ended up with five counties. In total, we sampled 27 counties in five provinces.

All townships in each of these 27 selected counties were divided according to their per-capita grassland area, and one township was randomly selected from each tercile, which yields a total of 81 townships. One village was then randomly selected from the higher per-capita grassland area tercile and the other, from the lower tercile of each selected township. Finally, six households were randomly selected from each of the 162 sampled villages, which yields a sample of 972 households. The study area and sample distribution are shown in Supplementary Fig. 3. All the participants gave their consent to surveyors at the very beginning of the interview, and participants were informed that they could end the interview at any time for any reason. Given this survey structure, our universities did not require ethical approval.

Structured survey questionnaires were designed to elicit information by interviewing household heads for the data for the years of 2008–2010 and 2015−2017 in Qinghai and Gansu and 2016−2017 in Inner Mongolia, Xinjiang, and Tibet. First, annual household income per capita was calculated by first summing up annual household net pastoral income, annual non-pastoral income, and annual GECP subsidy and then dividing the sum by the number of members in a household. We obtained annual net pastoral income by first asking herders about their gross income from selling livestock and livestock byproducts (which, when added together, generate gross pastoral income), along with inquiring about livestock production costs, and then subtracting pastoral production costs from gross pastoral income. We obtained gross non-pastoral income by first asking each family member whether he or she had any non-pastoral jobs and, if that family member did, asking how much money in total he or she earned from those jobs.

Second, subsidy intensity was calculated by dividing the annual GECP subsidy by operated farm size. The exact amount of money received from the GECP program by each herder household was obtained by asking herders how much money they received in each of the post-program years. We assigned a value of zero to GECP subsidies for all of the years before the program started.

Third, we calculated the total year-end livestock per household and supplementary feeding, using data reported by the respondents. This was done by first counting the year-end total headcount of animals of each different type (e.g., cattle, sheep, and horses). The different headcounts were then transformed into standardized sheep units, which are based on the relative feed intakes of different livestock. The year-end headcounts of animals of each type were reported by

household heads. The amount of supplementary feeding was measured on all feedstuffs that did not come from grazing. Information on whether a herder household rented in grassland, that is, whether a household paid other households to graze their land, was also directly reported by the respondents.

We also collected a number of household characteristics as control variables. To elicit this information, the survey team asked household heads questions about the quantity of labor used in raising livestock, years of education of each labor, operated farm size (in hectares), share of the jointly operated area (jointly operated land that a herder uses/total operated farm size), and the total number of different plots that comprised their grassland area. Grass harvesting is defined as a dummy variable equal to 1 if the herder harvested grass for supplementary feeding, and 0 otherwise. Crop/fodder planting is a dummy variable equal to 1 if the herder planted crop or fodder as supplementary feeding, and 0 otherwise.

*Other measures.* Other information at village and township levels also was collected by the survey team. Village leaders were asked whether their village had any local policy measures in place to limit grazing intensity. For example, some village authorities have been known to try to limit the total number of sheep units that herders can graze in an area by the use of local rules or norms. Village leaders also were asked whether their village had a formal government-run monitoring system on grazing intensity. For example, in some villages, government-hired staff were paid to monitor herds' grazing behavior. Farm-gate livestock prices, hay prices, wages for non-pastoral employment, and grassland rental prices were obtained from township accounting offices. Livestock price was calculated as a weighted average of the prices of all types of livestock, with their year-end counts as weights. Descriptive statistics of our key variables are shown in Supplementary Table 8.

**Empirical methods.** To assess the impacts of GECP on grassland quality, we use two types of approaches: a DID method applied to county-level data and an FE model applied to household-level data. To assess the impacts on herder income and equity, we, again, apply an FE model to household-level data. In this section, we first describe the DID method and then the FE model. Data analysis was performed in Stata 15.

*DID method.* In the DID approach, identification of GECP impacts comes from the year-to-year change in grassland quality following the introduction of GECP in the treated counties, compared to the contemporaneous change in the control counties. The treatment group includes all counties in five North and Northwestern program provinces that were covered in GECP-I in 2011–2015 (i.e., Xinjiang, Qinghai, Gansu, Ningxia, and Inner Mongolia). The control group includes all counties in five North and Northeastern provinces that were not covered by GECP-I until 2016 (i.e., Shanxi, Hebei, Liaoning, Jilin, and Heilongjiang).

The DID method uses the following specification:

$$Y_{lt} = \alpha_l + \beta(P_l \times T_t) + \mathbf{X}'_{lt}\gamma + \delta_t + \varepsilon_{lt}. \tag{1}$$

In this equation, $Y_{lt}$ is the grassland quality of county $l$, measured by the logarithm of its NDVI, recorded in a relevant time period $t$. A county $l$'s treatment status is denoted by a binary indicator $P$: $P_l = 1$ if county $l$ is a program county covered in GECP-I in 2011–2015 (treatment group), and $P_l = 0$ otherwise (control group). The time period is denoted by $T$: $T = 0$ for periods before GECP-I was implemented (2001–2010) and $T = 1$ for periods after (2011–2015). The coefficient $\beta$ measures the average treatment effect on the treated counties, which represents the average change in grassland quality in the treatment group relative to that of the control group. The vector $\mathbf{X}_{lt}$ includes a set of county-level climate and socioeconomic factors (Supplementary Table 5). We also control for time fixed effects ($\delta_t$), accounting for the fluctuations over the years common to all counties. Next, $\alpha_l$ denotes county fixed effects (county-specific constant terms), which control for permanent differences in grassland quality across localities.

In addition to the main model, we also analyze the potential mechanisms by which GECP affects grassland quality. Herders can respond to GECP payments in many ways, such as reducing grazing scale, changing livestock structure, increasing supplementary feeding, or enlarging operated grassland areas by renting in more grassland. Unfortunately, because we have only county-level data for year-end cattle and sheep inventories, the DID mechanism analysis at the county level examines only changes in cattle and sheep production. To track these changes, we replace the dependent variable in Eq. (1) with the logarithms of year-end cattle and year-end sheep inventory, as they are the most common livestock types.

Note that, because our estimation involves multiple years for the same set of counties, in all DID analyses, we adjust standard errors to be clustered at the county level to address potential within-county error autocorrelation over time[36]. Such a clustered variance-covariance estimator allows for non-parametric error correlations within clusters (counties) when the number of clusters is large[37], which is the case for our study.

To test the parallel assumption, we formally estimate the following event study specification:

$$Y_{lt} = \alpha_l + \sum_{k=-5}^{-1}\beta_0 1(k=t) + \sum_{m=1}^{5}\beta_1 1(m=t) + \mathbf{X}'_{lt}\gamma + \delta_t + \varepsilon_{lt}, \tag{2}$$

where the $\beta_0$ terms are the coefficients on the dummy variables for each of the pre-GECP years (2006–2010) and the $\beta_1$ terms are the coefficients on the dummy

variables for each year after GECP implementation (2011–2015). These coefficients measure changes in grassland quality relative to the level in 2011, the reference year.

**FE model.** We apply FE models to household-level data to estimate the impacts of GECP on both grassland quality and herder income. Letting $i$ denote households, $j$ for villages, $k$ for townships, $l$ for counties, and $t$ for years, we first specify the following model:

$$Y_{ijklt} = \alpha_{ijkl} + \beta P_{ijklt} + \mathbf{H}'_{ijklt}\boldsymbol{\gamma} + \mathbf{V}'_{jklt}\boldsymbol{\delta} + \mathbf{T}'_{klt}\boldsymbol{\theta} + \mu_{ijkl} + \tau_t + \varepsilon_{ijklt}. \quad (3)$$

When evaluating grassland quality, $Y_{ijklt}$ represents the logarithm of NDVI of household $i$ in year $t$. The key explanatory variable $P_{ijklt}$ is payment intensity measured by GECP payment per hectare (in logarithm) received by household $i$ in year $t$. $\mathbf{H}_{ijklt}$ is a vector of household characteristics, including operated farm size, number of livestock workers, number of farmland plots, share of the joint operated area, a grass harvest indicator, and a crop/harvest indicator. $\mathbf{V}_{jklt}$ is a vector of village characteristics, including an indicator of whether a village has local grassroots policy measures in place to limit grazing intensity, an indicator of whether a village has a formal government-run monitoring system, and village-level climate variables. Village-level climate variables include the cumulative rainfall and the mean temperature from May to October in each year. $\mathbf{T}_{klt}$ is a vector of township characteristics, including farm-gate livestock price, hay price, the wage for non-pastoral employment, and grassland rental price. $\mu_{ijkl}$ represents household fixed effects that capture the influence of time-invariant household characteristics, such as gender and education. $\tau_t$ represents year fixed effects that capture the time trend common to all counties. $\varepsilon_{ijklt}$ is the error term clustered at the village-by-year level. $\beta$, the coefficient of the key explanatory variable ($P_{ijklt}$), captures the impacts of GECP payment intensity.

When evaluating herder income, $Y_{ijklt}$ represents household income per capita, transformed using an inverse hyperbolic sine function, for household $i$ that resides in village $j$ in township $k$ in county $l$ in year $t$. We also estimate the impacts of GECP on non-GECP components of household income. In those analyses, $Y_{ijklt}$ represents net pastoral income per capita, non-pastoral income per capita, and non-GECP income per capita in an inverse hyperbolic sine transformation. $P_{ijklt}$ represents annual GECP payment per household in an inverse hyperbolic sine transformation. All other specifications are the same as those for evaluating the impact of GECP payments on grassland quality.

The rich information in the household-level data also allows us to explore some mechanisms through which GECP affects grassland quality. As stated earlier, herders may reduce grazing scale, change livestock structure, increase supplementary feeding, or enlarge operated grassland areas in response to GECP payments. When Eq. (3) is used to analyze the mechanism, $Y_{ijklt}$ may represent the number of year-end livestock in sheep units, number of year-end sheep, number of year-end cattle, supplementary feeding in kg per sheep unit, and whether a household rents in grassland. All of the explanatory variables are the same as those used in the evaluation of grassland quality.

To rule out reverse causality from the outcome variables to our key explanatory variable (payment intensity), we conduct an event study using the following specification:

$$Y_{ijklt} = \alpha_{ijkl} + \sum_{m=-3}^{-1} \beta_0 (\bar{P}_{ijklT} \times D_m) + \sum_{n=5}^{7} \beta_1 (\bar{P}_{ijklT} \times D_n) + \mathbf{H}'_{ijklt}\boldsymbol{\gamma} + \mathbf{V}'_{jklt}\boldsymbol{\delta} + \mathbf{T}'_{klt}\boldsymbol{\theta} + \mu_{ijkl} + \tau_t + \varepsilon_{ijklt}, \quad (4)$$

where the $\beta_0$ terms are the coefficients on a set of interaction terms, defined as the dummy variables ($D_m$) for each of the pre-GECP years (2008–2010) multiplied by the average payment intensity ($\bar{P}_{ijklT}$) during the period of 2015–2017. Similarly, the $\beta_1$ terms are the coefficients on a set of interaction terms, defined as the dummy variables for each year after GECP implementation (2015–2017) ($D_n$) multiplied by the average payment intensity during the period 2015–2017 ($\bar{P}_{ijklT}$). The reference year is 2010. Given this setup, a small and statistically insignificant estimate of $\beta_0$ will lend support to the plausibility of the pre-trend assumption.

We also use an FE model to examine the heterogeneous effects of GECP at the household level. Specifically, we examine how GECP's impact varies with different levels of grassland area per capita, non-pastoral wage, grassroots measures to limit grazing intensity, and a formal system to monitor grazing intensity. To estimate these heterogeneous effects, we add an interaction term ($P_{ijklt} \cdot X_{ijklt}$) in Eq. (3) as follows:

$$Y_{ijklt} = \gamma_0 + \gamma_1 P_{ijklt} + \gamma_2 P_{ijklt} \cdot X_{ijklt} + \mathbf{H}'_{ijklt}\boldsymbol{\gamma_3} + \mathbf{V}'_{jklt}\boldsymbol{\gamma_4} + \mathbf{T}'_{klt}\boldsymbol{\gamma_5} + \varphi_{ijkl} + \gamma_t + \varepsilon_{ijklt}, \quad (5)$$

where $X_{ijklt}$ represents subgroups of interest, defined by different values of (i) per-capita grassland area operated by household $i$, (ii) township-level wage for non-pastoral employment, (iii) whether the household's village has local grassroots measures in place to limit grazing intensity, or (iv) whether a village has formally monitored grazing intensity.

To further examine the equity effect of GECP on herder income (and, thus, fulfill our third objective), we add a set of interaction terms in Eq. (3) as follows:

$$Y_{ijklt} = \delta_1 P_{ijklt} \cdot IL_{ijkl} + \delta_2 P_{ijklt} \cdot IM_{ijkl} + \delta_3 P_{ijklt} \cdot IH_{ijkl} + \mathbf{H}'_{ijklt}\boldsymbol{\delta_4} + \mathbf{V}'_{jklt}\boldsymbol{\delta_5} + \mathbf{T}'_{klt}\boldsymbol{\delta_6} + \tau_t + \varepsilon_{ijklt}. \quad (6)$$

Specifically, we divide the sample into three terciles, i.e., low-, medium-, and high-income groups, according to household income in 2010. $IL$ ($IM$ or $IH$) is equal to 1 if the household belongs to the low- (medium- or high-) income group, and 0 otherwise. $\delta_1$ to $\delta_3$ are the coefficients of primary interest, which indicate the GECP impact on income for different groups. All other variables are defined in the same way as in Eq. (3). Note finally that, to address potential within-village error autocorrelation over time, we adjust the standard errors to be clustered at the village-by-year level.

## Data availability

The data that support the findings of this study are available from Harvard Dataverse: https://doi.org/10.7910/DVN/IWLO8T.

## Code availability

The *Stata* code used for the main analysis of this study is available from Harvard Dataverse: https://doi.org/10.7910/DVN/IWLO8T.

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

## Acknowledgements

This work acknowledges the financial support from the National Natural Science Foundation of China (Grant no. 71773003 and 71934003), the Chinese Academy of Engineering (Grant no. 2018-XZ-25; 2020-XZ-29), the Fundamental Research Funds for the Central Universities (Grant no. lzujbky-2019-kb28; lzujbky-2020-kb29) and the National Social Science Foundation of China (Grant no. 18ZDA074). We thank Qisheng Feng and Dongqing Li for their excellent research assistant. We thank Eric Wang and Sharon Bear for their excellent editing work. We also thank Quentin Grafton, Michael Delgado, Qiran Zhao and participants of the 2018 CAER-IFPRI annual conference in Guangzhou, for their helpful comments on an earlier version of this paper.

## Author contributions

L.H. contributes to the conception and design of the work. Y.H., L.H., and J.H. contribute to collecting and preparing data. L.H., F.X., Q.C., and Y.H. contribute to data analysis. L.H., F.X., Q.C., and Y.H. have drafted the work. J.H., N.R., and S.R. substantively revised it.

## Competing interests

The authors declare no competing interests.
