## [Peer Review File · Nature Communications]

Reviewer comments, first round -

Reviewer #1 (Remarks to the Author):

This study evaluated the impacts of grassland ecological compensation policy on grassland quality and income based on remote sensing and herder survey. It is an interesting topic. But in China, grassland covered a wide area and has a spatial heterogeneous in grassland type, herdsman's living style etc. from the east to the west, from the north to the south. The research conducted surveys only in two provinces of Qinghai and Gansu, the result is not convinced or typical for China. I suggested more survey should be supplied in Inner Mongolia grassland. Also, for the impacts of GECP on grassland and herders, some spatial characteristic should be further explored. For example, did the grassland quality increase induced by the policy have spatial difference? How drive the difference? Did the Inequality have spatial heterogeneity? Why? In addition, the writing of the paper was not good enough because of the weak logic. And the language was not concise enough.

Referee report on
“Grassland Ecological Compensation Policy in China:
An Evaluation of Ecological, Income, and Inequality Effects”

This paper evaluates the ecological and economic impacts of China’s Grassland Ecological Compensation Policy (GECP), one of the largest PES programs in the world. The authors assemble an impressive set of data from administrative sources and from surveys, and adopt the difference-in-difference (DID) method to establish a causal relationship between GECP and the outcome variables. They provide solid evidence that GECP moderately improved grassland quality, raised income of recipients, but also increased income inequality with richer herders receiving more payments. The paper explores not only the impacts but also the mechanisms through which the impacts materialize, showing that herders reduced the inventory of sheep but not of cattle. It draws a convincing set of lessons for program design that are relevant not only for GECP but also for large scale PES programs in general.

The paper’s topic is extremely important, not only because of the sheer size of GECP but also because of the many challenges facing PES program designs worldwide. There are few systematic studies of major PES programs in developing countries and particularly in China, and a major contribution of this paper is to adopt the state of the art program evaluation methodologies such as DID and apply them to a carefully assembled data set to establish causality. Methodological soundness is critical for large program evaluations because these programs tend to have many confounding factors, so that correlation often does not imply causality, and causal interpretations of correlation can lead to false conclusions and recommendations about program design. By being careful about causality, the paper sets a good example for interdisciplinary research approaches in dealing with confounding factors.

I will start my comments about the Methods section, mainly on the econometric models.

- 1) Some comments on semantics: DID is an estimation method or technique, but not a model. FE (fixed effect) is indeed a model.
- 2) The survey solicits recall data that go back many years: while the survey was held in 2017-18, the earliest data surveyed are about detailed decisions and incomes back to 2008. One would expect significant measurement errors. Since the household data are analyzed using FE models, you might have tremendous attenuation bias. This might partly explain your finding that GECP payments’ impacts are small. I don’t believe that there are good ways to overcome this problem, but you should add some comments about this issue in the paper. You could also

estimate the model using RE, and see whether the estimates are much different from those of FE.

- 3) The DID model in equation (1) is not entirely correct. Since you already control for the county and year FE, the two variables P_t and T_t will be absorbed by these FE. A standard way to implement DID in a FE model is to remove these two terms (but keep their interaction terms). The reason you were able to implement (1), if I understand your results correctly, is that you included province FE instead of county FE. I can't tell this in Table 1 since the table is very parsimonious but this seems to be the case in Tables S2 – S4. I would suggest that you run a standard FE model to implement DID.
- 4) Also in (1), why are the covariates X included at their year 2010 values? A natural way to include them is to have X_{it} , especially for the weather variables. Why would one expect that the weather condition in year 2010 affect the grassland quality in 2015?
- 5) For income, a more standard transformation is the log transformation. Is there any particular reason to use the arcsinh transformation? You shouldn't have any observations of zero income. Would your results change a lot if you use log transformation?
- 6) Some of the results are hard to understand without a formal econometric model written down for it. For example, in column (1) of Table S3, you have both a year FE and a post-GECD time T. Since there is only one year before and after the program (2010 and 2011), then one of the three coefficients of 2010, 2011, and T is redundant in the estimation. Then what does the coefficient of T capture? This case also highlights a point I made earlier: it is much cleaner to use FE to implement DID.

Comments about the Results section

- 7) The program started to be implemented in 2011, and one would expect some lag for it to lead to significant changes in grassland quality. Your main DID result in Table 1 column (1) is based on two years before (2008-10) and two years after (2011-13). Why would one expect the impacts of the program to show up within two years? In the robustness check results of Table S3, the DID estimate is rather stable as you expand the time horizon to 2006-10 vs 2011-15. Could you explain why the estimated impacts are stable over time instead of increasing over time? One reason might be that, as you extend the time frame, you are also increasing the years included before the program, and those years might have good weather conditions that improved the grassland quality – recall that you fixed the weather conditions in 2010 rather than allowing them to vary across years. To look at the time trend, I suggest that you (i) include X_{it} rather than

X_{it_0} , (ii) implement DID on two years before and after the program, with 2010 being the before year, and vary the after year from 2011 to 2015 (i.e., estimate 5 models), and see whether the estimated DID coefficient increases or not.

- 8) In Table 1 columns (2) and (3), FE and RE are estimated. Are the FE/RE at the county level? Or at provincial level? Note that a county level FE model should be the same as the DID model you estimated in column (1). Again, without presenting or clarifying the specific estimation model, the results are sometimes hard to interpret.
- 9) You used three year averages to test the parallel trend assumption. Since your analysis is conducted using annual data, a more convincing approach is to do an event analysis using annual data, and see whether the coefficients of being treated are significant before 2010. Another “standard” approach is to graphically present the average values of the outcome variables for the treated and control groups, or a residualized plot with some minimal set of control variables.
- 10) I had a hard time understanding how you conducted the common trend test in Table S5 without seeing an estimation equation. For example, in column (1), how does “payment intensity” vary across years? Suppose you used $PaymentIntensity_{it}$, then is the intensity equal to zero before 2010? Or you used a certain average of the payment intensity? Again, you could simply conduct an event analysis to establish common trend.
- 11) You provided convincing evidence that richer households received more GECP payments. But why? What are the criteria used in deciding who receives how much? You should include in the paper at least a couple of graphs describing in more details important program design features of GECP, so that one will have an easier time to understand your estimation results.
- 12) Since GECP is designed to alter herder behavior, and since you do observe behavioral changes (e.g., reduced stock of sheep), one would expect that non-program income (i.e., total household income minus GECP payments) will change. Theoretically the non-program income is likely to decrease (that’s why external payment is needed to induce non-voluntary behavioral changes), but a well designed program in fact can lead to income increases. I strongly suggest that, in addition to looking at the program’s effects on total income, you also look at the program’s effects on non-program income.
- 13) One effect you identified is that herders rented more land in response to the program. Where did the rental land come from? Is it grassland from other herders? Or from other kinds of land

converted to grassland? Did the program lead to more or less grassland in total? These questions are important in understanding the overall ecological impacts of GECP.

Comments about the presentation of the paper.

- 14) You should mention early on that the treatment and control counties are border counties in the 2011-2015 and post-2016 provinces. Currently these are mentioned in the Data sub-section of the Methods section.
- 15) Readers of the DID literature often are interested in the comparisons of the summary statistics between the treated and control groups. Table S6 are about the provinces of the different phases, which include not only the provincial border counties. You should extend this table or have a separate table that shows the comparisons.

Response to Reviewer #1

Comment 1: This study evaluated the impacts of grassland ecological compensation policy on grassland quality and income based on remote sensing and herder survey. It is an interesting topic. But in China, grassland covered a wide area and has a spatial heterogeneous in grassland type, herdsman's living style etc. from the east to the west, from the north to the south. The research conducted surveys only in two provinces of Qinghai and Gansu, the result is not convinced or typical for China. I suggested more survey should be supplied in Inner Mongolia grassland. Also, for the impacts of GECP on grassland and herders, some spatial characteristic should be further explored. For example, did the grassland quality increase induced by the policy have spatial difference? How drive the difference? Did the Inequality have spatial heterogeneity? Why?

Response: We thank the reviewer for their valuable suggestions. We agree that China's grassland covers a wide area and has spatially heterogeneous characteristics. Although we have already provided some heterogeneity analyses in the previous version of the manuscript on page 11, we did not focus on *spatial* heterogeneity. Following the reviewer's suggestion, we have added results of further explorations on spatial heterogeneity in the revised manuscript.

To analyze spatial heterogeneity at the county level, we used grassland type as an indicator under the assumption that different grassland types reflect different natural resource endowments and affect the nature of each region's economic activities. Following Ma and Xu (2019), we adopted the classification of 5 grassland types (i.e. meadow, grassland, desert, shrubland, and herbosa). Our county-level dataset covers all 5 types (see Figure R1 below). Since the household level data only covers a subset of these, and given the small sample sizes that would result from doing the subanalysis at the household level, we focus our efforts on the county-level analysis.

Figure R1. Map of grassland types in study area

After analyzing the spatially heterogeneous impacts of the GECP on grassland quality at the county level, we added Panel C to Table 6 in the revised manuscript. According to the newly added results, the GECP impacts on improving grassland quality are positive and statistically significant in four out of five grassland types (grasslands, desert, shrubland and herbosa, but not meadows). In other words, we found that the GECP program raised grassland quality in all areas of China (including counties that are classified as having primarily grasslands, desert, shrubland and herbosa) with the exception of counties that are classified as having primarily meadows. The coefficients of the GECP program variable vary across the four areas where statistically significant impacts were found (from 0.018 for shrubland to 0.055 for grassland). The coefficient on the GECP program variable for counties primarily with meadows is also positive, although the point estimate is small (and, as stated, is statistically insignificant). The results suggested that even though there may exist some spatial heterogeneity by the natural resource endowment of counties, the results are mostly consistent across space: The GECP program has had a positive effect on grassland quality in most counties (classified as having different types of grassland).

In the revised manuscript, we have added a paragraph in the “Results” section (pages 12 to 13), which reads:

“Third, we explored potential spatial heterogeneity as related to the impact of GECP. We use grassland type as an indicator for spatial heterogeneity, under the assumption that different types of grassland reflect different natural resource endowments and affect the nature of each region’s economic activities. Following Ma and Xu (2019), we adopt the classification of five grassland types (i.e., meadow, grassland, desert, shrubland, and herbosa), which are all covered in our county-level data.

The results (Table 6, Panel C) suggest that, although there exists some spatial heterogeneity by the nature of the resource endowments of counties, the effects are consistent mainly across space: The GECP program has had a positive effect on grassland quality in most counties (classified as having different types of grassland) in the program area. More specifically, the GECP program raised grassland quality in all counties (including counties classified as having primarily grassland, desert, shrubland, and herbosa), with the exception of counties that are classified as having primarily meadows. The coefficients for the GECP program variable vary across the four areas in which statistically significant impacts were found (from 0.018 for shrubland to 0.055 for grassland). The coefficient for the GECP program variable for counties with meadows also is positive, although the point estimate is small (and, as stated, statistically insignificant). ”

Table 6. Estimated heterogeneous impacts of GECP on grassland quality: county level

	(1)	(2)	(3)		
Panel A by rural road intensity in 2008:	Low	Medium	High		
$P \times T$	0.017	0.025***	0.053***		
	(0.011)	(0.009)	(0.010)		
No. of observations	1,122	1,144	1,149		
R^2	0.994	0.988	0.977		
Panel B by NDVI in 2008:	Low	Medium	High		
$P \times T$	0.035***	0.055***	0.026***		
	(0.009)	(0.009)	(0.008)		
No. of observations	1,128	1,150	1,086		
R^2	0.991	0.966	0.948		
Panel C by grassland type:	Meadow	Grassland	Desert	Shrub land	Herbosa
$P \times T$	0.001	0.055***	0.030***	0.018**	0.038***
	(0.006)	(0.008)	(0.009)	(0.009)	(0.007)
No. of observations	2,038	2,380	2,231	1,894	1,714
R^2	0.969	0.979	0.995	0.952	0.953

Note: The dependent variable is NDVI in log form. In Panels A and B, we present the results when we first divide the control and treatment groups into three terciles (i.e., low, medium, and high), based on each indicator in a base year (i.e., 2008). In Panel C, all of the control and treatment counties are grouped into five grassland types. We then pair the subgroups in the control and treatment groups and run the model indicated by Equation (1). All other model specifications are the same as in Column (1) of Table 1.

* $p < 0.10$, ** $p < 0.05$, *** $p < 0.01$

The reviewer also suggested that we expand our analysis to Inner Mongolia. We agree with the reviewer that including data from Inner Mongolia would have strengthened our analysis, and in designing the study, we did try to include data from that province. But the data on Inner Mongolia are incomplete for analyzing the impacts of the GECP, as we do not have the data for the pre-program period. Even if we wanted to return to Inner Mongolia now and expand the survey, the current policy regime (post-COVID-19 travel restrictions) prevents us from doing so.

Fortunately, previous studies have examined similar issues in Inner Mongolia. For example, Hu et al. (2019) conducted a survey in Inner Mongolia in 2015 and presented the results for the impact of the GECP (on livestock production) in their paper. Although their study did not examine the impacts of the GECP on grassland quality directly, their results on the impacts on livestock production are quite consistent with our findings in Qinghai and Gansu. Hence, in the “Discussion” section of the revised manuscript we have included a new paragraph that summarizes their findings (page 26), to facilitate comparisons of their results with ours based on data from Qinghai and Gansu.

We make this point in the revised “Discussion” section on page 26 by writing:

“Herders respond to GECP payments through behavioral changes, specifically, by reducing herd sizes and increasing the scale of their farm operations. Our results show that GECP has been able to improve grassland quality by reducing the size of sheep herds (although we did not find any reductions in the size of cattle herds). In fact, these findings are consistent with the results found in Hu et al. (2019). In their survey in Inner Mongolia, the research team found that GECP payments prompted farmers to reduce sheep herds but not cattle herds. These findings are also supported by our field observations, which showed that herders believe that raising sheep causes more damage to the grassland than raising cattle due to their different feeding and trampling habits. We also found that herders tended to expand their farm size by renting in grassland to reduce grazing intensity. In other words, farm sizes increased through renting in grassland from other herders. Then, as herders rented out their grassland, their labor was (at least partially) released from the pastoral sector, allowing them to take on non-pastoral jobs. Therefore, it is through the non-pastoral jobs that the pressure on the grassland was reduced, as fewer households were relying on the grassland to make a living. Interestingly, herders did not use more supplementary feeding, which may reflect the incompleteness of hay markets or constraints on cash liquidity.”

References:

- Ma, K., & Xu, Z. China National Specimen Information Infrastructure. Chinese Academy of Sciences (CAS). Metadata dataset <https://doi.org/10.15468/kmob80> accessed via GBIF.org on 2020-07-20 (2019).
- Hu, Y., Huang, J. & Hou, L. Impacts of the grassland ecological compensation policy on household livestock production in China: an empirical study in Inner Mongolia. *Ecol. Econ.* 161, 248–256 (2019).

Comment 2: In addition, the writing of the paper was not good enough because of the weak logic. And the language was not concise enough.

Response: We thank the reviewer for pointing this out. During the revision process, the authors have worked closely and carefully to improve the logic of the paper. We also employed a deeply experienced editor to polish the paper again. We believe the paper reads more fluently now.

Response to Reviewer #2

This paper evaluates the ecological and economic impacts of China's Grassland Ecological Compensation Policy (GECP), one of the largest PES programs in the world. The authors assemble an impressive set of data from administrative sources and from surveys, and adopt the difference-in-difference (DID) method to establish a causal relationship between GECP and the outcome variables. They provide solid evidence that GECP moderately improved grassland quality, raised income of recipients, but also increased income inequality with richer herders receiving more payments. The paper explores not only the impacts but also the mechanisms through which the impacts materialize, showing that herders reduced the inventory of sheep but not of cattle. It draws a convincing set of lessons for program design that are relevant not only for GECP but also for large scale PES programs in general. The paper's topic is extremely important, not only because of the sheer size of GECP but also because of the many challenges facing PES program designs worldwide. There are few systematic studies of major PES programs in developing countries and particularly in China, and a major contribution of this paper is to adopt the state of the art program evaluation methodologies such as DID and apply them to a carefully assembled data set to establish causality. Methodological soundness is critical for large program evaluations because these programs tend to have many confounding factors, so that correlation often does not imply causality, and causal interpretations of correlation can lead to false conclusions and recommendations about program design. By being careful about causality, the paper sets a good example for interdisciplinary research approaches in dealing with confounding factors.

Comment 1: Some comments on semantics: DID is an estimation method or technique, but not a model. FE (fixed effect) is indeed a model.

Response: We thank the reviewer for pointing this out. We have corrected this throughout the manuscript.

Comment 2: The survey solicits recall data that go back many years: while the survey was held in 2017-18, the earliest data surveyed are about detailed decisions and incomes back to 2008. One would expect significant measurement errors. Since the household data are analyzed using FE models, you might have tremendous attenuation bias. This might partly explain your finding that GECP payments' impacts are small. I don't believe that there are good ways to overcome this problem, but you should add some comments about this issue in the paper. You could also estimate the model using RE, and see whether the estimates are much different from those of FE.

Response: We thank the reviewer for pointing out the potential measurement error and attenuation bias issues. We agree with the reviewer that some of our variables (such as herder income, which is used to create a set of dependent variables) may suffer from recall bias. However, not all of our variables suffer from this bias. The data for grassland quality (another dependent variable) are measured from remote sensing images (NDVI), and are thus not subject to measurement errors stemming from recall bias. Additionally, GECP payment data (our key independent variable) come from herder bank records (provided by the herders themselves during the survey), which are similarly insulated from recall bias. While the reviewer is correct in pointing out that measurement error

(and thus attenuation bias) may exist in our data, not all of our variables suffer from such measurement error (for example, random measurement errors in the dependent variable will not lead to attenuation bias; it simply increases the variance of the error term, making the standard errors of the estimated coefficients larger).

Additionally, following the reviewer’s suggestion, we ran both FE and RE models as robustness checks against the attenuation bias in the income data. The results are shown in Table S9 below. The results from analysis using the RE model are quite comparable to those from FE models, although the magnitude of the coefficients in the net pastoral income and non-pastoral income models are slightly different. The FE models estimated that a 10% increase in annual GECP payments led to a 1.67% increase in non-pastoral income per capita (Col. 5), while the RE models estimated that a 10% increase in annual GECP payments led to a 3.46% increase in non-pastoral income per capita (Col. 6) . As the reviewer has mentioned, FE models suffer more from attenuation bias (if it exists) than RE models; however, RE models may suffer more from endogeneity issues than FE models. Therefore, while comparing FE and RE results cannot definitely pin down the existence of attenuation bias, it does serve as a good robustness check. In any case, both models point to a positive impact of the GECP program, which should build confidence in the results.

Table S9. Estimated impacts of the GECP on herder income: household level

	(1)	(2)	(3)	(4)	(5)	(6)
	Household income per capita		Net pastoral income per capita		Non-pastoral income per capita	
	FE	RE	FE	RE	FE	RE
Annual GECP payment (yuan)	0.254*** (0.096)	0.249*** (0.060)	0.062 (0.114)	0.128 (0.080)	0.167** (0.078)	0.346*** (0.078)
Control variables	Yes	Yes	Yes	Yes	Yes	Yes
Year fixed effects	Yes	Yes	Yes	Yes	Yes	Yes
Household fixed effects	Yes	No	Yes	No	Yes	No
No. of observations	2,024	2,024	2,024	2,024	2,024	2,024
Overall R^2	0.656	0.323	0.679	0.251	0.828	0.474

Note: All dependent variables and the key independent variable, annual GECP payment, are transformed using an inverse hyperbolic sine transformation to avoid taking logarithm of zero numbers, following $\ln(y+(y^2+1)^{1/2})$. Household income includes net pastoral income, non-pastoral income, and GECP payment. Household-, village-, and township-level time-variant

variables are controlled for. Household-level controls include quantity of labor used in raising livestock, operated farm size, share of joint operated area, total number of different plots, a dummy variable for grassland harvesting, and a dummy variable for planting crop/fodder. Village-level controls include an indicator of whether a village has local grassroots measures in place to limit grazing intensity and an indicator of whether a village has a formal government-run monitoring system. Township-level controls include farm-gate livestock prices, hay prices, wages for non-pastoral employment, and grassland rental prices. County-level climate variables include cumulative rainfall and mean temperature from May to October in each year. Standard errors are in parentheses, clustered by village and year.

* $p < 0.10$, ** $p < 0.05$, *** $p < 0.01$

Since these robustness checks all confirm our FE model findings, we report the FE estimates alone in the main manuscript (Table 2). We have, however, acknowledged this issue in the Discussion by adding the following section on Page 29:

“Although this study certainly provides important contributions to the literature, it has certain limitations. We acknowledge that our income data may suffer from measurement error, as our income variable is constructed based on recall data. Fortunately, not all of our variables rely on recall data: Variables that measure grassland quality are based on remote sensing images, and GECP payments are based on herder bank records, neither of which relies on recall data, and are, therefore, less likely to suffer from measurement error. In addition, we estimated both FE and RE models as robustness checks against attenuation bias for income data (as RE models would suffer less from attenuation bias, if it exists). The results from the RE models are presented in Table S9 and are mostly consistent with the results from our FE models.”

Comment 3: The DID model in equation (1) is not entirely correct. Since you already control for the county and year FE, the two variables $P P l l$ and $T T t t$ will be absorbed by these FE. A standard way to implement DID in a FE model is to remove these two terms (but keep their interaction terms). The reason you were able to implement (1), if I understand your results correctly, is that you included province FE instead of county FE. I can’t tell this in Table 1 since the table is very parsimonious but this seems to be the case in Tables S2 – S4. I would suggest that you run a standard FE model to implement DID. Also in [equation] (1), why are the covariates X included at their year 2010 values? A natural way to include them is to have X_{lt} , especially for the weather variables. Why would one expect that the weather condition in year 2010 affect the grassland quality in 2015?

Response: We agree with these points and have rewritten equation (1) as follows:

$$Y_{lt} = \alpha_l + \beta(P_l \times T_t) + \mathbf{X}'_{lt}\boldsymbol{\gamma} + \delta_t + \varepsilon_{lt}. \quad (1)$$

In this equation, Y_{lt} is the grassland quality of county l , measured by the logarithm of its NDVI, recorded over time period t . A county l 's program participation status is denoted

by a binary indicator P : $P_l = 1$ if county l is a program county covered in the GECP-I in 2011–2015 (treatment group), and $P_l = 0$ otherwise (control group). The time period is denoted by T : $T = 0$ for periods before the GECP-I was implemented (2001–2010); $T = 1$ for periods after (2011–2015). The vector \mathbf{X}_{lt} includes a set of county-level climate and socioeconomic factors (Table S6). The coefficient β identifies the average treatment effect on the treated, which represents the average difference in grassland quality in the treatment group relative to the control group. We also control for time fixed effects (δ_t), which account for the fluctuations over the years common to all counties.

Further following the reviewer’s suggestion, we also controlled for the covariates \mathbf{X} at county l and year t , instead of using the baseline year (i.e. 2010) value in the revised manuscript. We have additionally changed our FE model to include county-level fixed effects instead of the previously used province-level FE.

After making all of these changes, we found that all the results are consistent with the previous findings (using with the county-level FE model and the province-level FE model).

To make model specifications clearer to readers, we have also specified the equation used (via equation numbers) for each column of the tables in the footnote of the table.

Comment 4: For income, a more standard transformation is the log transformation. Is there any particular reason to use the arcsinh transformation? You shouldn’t have any observations of zero income. Would your results change a lot if you use log transformation?

Response: We use the arcshinh transformation for the income variables because some households in our sample do not have non-pastoral income (which is a component of total income); that is, while we have positive values of household income for all households, we do have zero values of non-pastoral income for some households. The main reason we used the arcshinh transformation is to make the transformation consistent for all three income measures (total household income, pastoral income, and non-pastoral income). However, to address the reviewer’s concern, we have re-run the model using a log transformation; following common practice, we replaced Y with $Y + 1$ if $Y = 0$, to avoid unnecessarily truncating the sample. The results from this transformation are consistent with those based on the arcshinh transformation (See Table 2b below). We append the results here for your review but do not include them in the manuscript to avoid cluttering the table with repetitive information. We do, however, add one note at the bottom of Table 2: “The results are consistent between an inverse hyperbolic sine transformation and a log transformation of the dependent variable in each column. The log transformation results are available upon request.”

Table 2b. Estimated impacts of the GECP on herder income: household level, FE

	Household income per capita		Net pastoral income per capita		Non-pastoral income per capita	
	Arcsinh (1)	Log(Y+1) (2)	Arcsinh (3)	Log(Y+1) (4)	Arcsinh (5)	Log(Y+1) (6)
Annual GECP payment (yuan)	0.254*** (0.096)	0.245*** (0.037)	0.062 (0.114)	0.062 (0.105)	0.167** (0.078)	0.153** (0.072)
Control variables	Yes	Yes	Yes	Yes	Yes	Yes
Household fixed effect	Yes	Yes	Yes	Yes	Yes	Yes
Year fixed effect	Yes	Yes	Yes	Yes	Yes	Yes
No. of observations	2,024	2,024	2,024	2,024	2,024	2,024
Overall R ²	0.656	0.656	0.679	0.614	0.828	0.788

Note: In columns (1), (3) and (5), all dependent variables and the key independent variable (annual GECP payment) are transformed using an inverse hyperbolic sine transformation to avoid taking logarithm of zero numbers, following the expression $\ln(y+(y^2+1)^{1/2})$. In columns (2), (4) and (6), all dependent variables and the key independent variable (annual GECP payment) are taken a log transformation. Household income includes net pastoral income, non-pastoral income, and GECP payment. Both household and year fixed effects are controlled for all models. Household-, village-, and township-level time-variant variables are also controlled. Household-level controls include quantity of labor used in raising livestock, operated farm size, share of joint operated area, total number of different plots, a dummy variable for grassland harvesting, and a dummy variable for planting crop/fodder. Village-level controls include an indicator of whether a village has local grassroots measures in place to limit grazing intensity and an indicator of whether a village has a formal government-run monitoring system. Township-level controls include farm-gate livestock prices, hay prices, wages for non-pastoral employment, and grassland rental prices. County-level climate variables include cumulative rainfall and mean temperature from May to September in each year. Standard errors are in parentheses, clustered by village and year.

* $p < 0.10$, ** $p < 0.05$, *** $p < 0.01$

Comment 5: Some of the results are hard to understand without a formal econometric model written down for it. For example, in column (1) of Table S3, you have both a year FE and a post-GECD time T. Since there is only one year before and after the program (2010 and 2011), then one of the three coefficients of 2010, 2011, and T is redundant in the estimation. Then what does the coefficient of T capture? This case also highlights a point I made earlier: it is much cleaner to use FE to implement DID.

Response: As we have responded above to Comment 3, we now use an FE model to implement the DID approach per the reviewer’s suggestion (see equation 1 in Comment 3). We present the results as a robustness check in Table S3 by using different time periods. We have updated Table S3 accordingly, as can be seen below:

Table S3. Robustness checks for DID estimates of log(NDVI) with different post-program time periods

	(1)	(2)	(3)	(4)	(5)
Pre-program period:	2010	2010	2010	2010	2010
Post-program period:	2011	2011–2012	2011–2013	2011–2014	2011–2015
$P \times T$	0.001 (0.005)	0.034*** (0.008)	0.037*** (0.007)	0.047*** (0.007)	0.044*** (0.007)
Year fixed effects	Yes	Yes	Yes	Yes	Yes
County fixed effects	Yes	Yes	Yes	Yes	Yes
Climate controls	Yes	Yes	Yes	Yes	Yes
Socioeconomic controls	Yes	Yes	Yes	Yes	Yes
N	1,138	1,709	2,278	2,845	3,415
R^2	0.997	0.991	0.991	0.991	0.991

Note: The treatment group ($P = 1$) includes the counties in five North and Northwestern program provinces that were covered in GECP-I, i.e., Xinjiang, Qinghai, Gansu, Ningxia, and Inner Mongolia. The control group ($P = 0$) includes the counties in five North and Northeastern provinces that were not covered by GECP-I until 2016, i.e., Shanxi, Hebei, Liaoning, Jilin, and Heilongjiang. Year and province fixed effects are controlled for. Climate controls include monthly rainfall, temperature, and PSDI for May to October in each year. Socioeconomic controls include per-capita county fiscal income. Robust standard errors are in parentheses, adjusted for clustering at the county level.

* $p < 0.10$, ** $p < 0.05$, *** $p < 0.01$

Comment 6: The program started to be implemented in 2011, and one would expect some lag for it to lead to significant changes in grassland quality. Your main DID result in Table 1 column (1) is based on two years before (2008-10) and two years after (2011-13). Why would one expect the impacts of the program to show up within two years? In the robustness check results of Table S3, the DID estimate is rather stable as you expand the time horizon to 2006-10 vs 2011-15.

Could you explain why the estimated impacts are stable over time instead of increasing over time? One reason might be that, as you extend the time frame, you are also increasing the years included before the program, and those years might have good weather conditions that improved the grassland quality – recall that you fixed the weather conditions in 2010 rather than allowing them to vary across years. To look at the time trend, I suggest that you (i) include X_{it} rather than X_{it0} , (ii) implement DID on two years before and after the program, with 2010 being the before year, and vary the after year from 2011 to 2015 (i.e., estimating 5 models), and see whether the estimated DID coefficient increases or not.

Response: We thank the reviewer for these insightful comments, and have incorporated these comments into our 5 models (specifically, using X_{it} rather than X_{it0} , and reestimated the five models in which the total after-program years vary from just 2011 to 2011-2015). We present the results of these 5 models in Table S3 below. As the reviewer expected, some lag in GECP impacts was found. The GECP started to have positive impacts on improving grassland quality starting two years after program implementation. The impact stabilized from the 3rd year to the 5th year. In fact, we observed during the in-the-field survey that, after receiving GECP payments, it generally took herders 1-2 years to change their livestock management plans/practices. This lag explains why we see significant and positive impacts from 2012 onward (1-2 years after program implementation). Importantly, however, the GECP policy design, after initial implementation, does not change. Thus, herder behavior also remains unchanged (aside from initial implementation) (Ministry of Agriculture and Rural Affairs of China [MARA], 2011).

We have included Table S3 in the supplementary information and added the following sentence in the revised manuscript: “Further explorations found that GECP started to have positive impacts on improving grassland quality starting two years after program implementation (Table S3)”

Table S3. Robustness checks for DID estimates of log(NDVI) with different post-program time periods

	(1)	(2)	(3)	(4)	(5)
Pre-program period:	2010	2010	2010	2010	2010
Post-program period:	2011	2011–2012	2011–2013	2011–2014	2011–2015
$P \times T$	0.001 (0.005)	0.034*** (0.008)	0.037*** (0.007)	0.047*** (0.007)	0.044*** (0.007)
Year fixed effects	Yes	Yes	Yes	Yes	Yes
County fixed effects	Yes	Yes	Yes	Yes	Yes
Climate controls	Yes	Yes	Yes	Yes	Yes
Socioeconomic controls	Yes	Yes	Yes	Yes	Yes
N	1,138	1,709	2,278	2,845	3,415
R^2	0.997	0.991	0.991	0.991	0.991

Note: The treatment group ($P = 1$) includes the counties in five North and Northwestern program provinces that were covered in GECP-I, i.e., Xinjiang, Qinghai, Gansu, Ningxia, and Inner Mongolia. The control group ($P = 0$) includes the counties in five North and Northeastern provinces that were not covered by GECP-I until 2016, i.e., Shanxi, Hebei, Liaoning, Jilin, and Heilongjiang. Year and province fixed effects are controlled for. Climate controls include monthly rainfall, temperature, and PSDI for May to October in each year. Socioeconomic controls include per-capita county fiscal income. Robust standard errors are in parentheses, adjusted for clustering at the county level.

* $p < 0.10$, ** $p < 0.05$, *** $p < 0.01$

Reference:

Ministry of Agriculture and Rural Affairs of China. Guidance on the implementation of the Grassland Ecological Compensation Policy in 2011 (2011).

Comment 7: In Table 1 columns (2) and (3), FE and RE are estimated. Are the FE/RE at the county level? Or at provincial level? Note that a county level FE model should be the same as the DID model you estimated in column (1). Again, without presenting or clarifying the specific estimation model, the results are sometimes hard to interpret.

Response: The FE/RE in the original manuscript were estimated at the county level. Therefore, the county-level FE model has the same results as the DID estimates with the updated model (i.e. equation (1) in Comment 3). We therefore dropped the FE column

and only kept the DID and RE column in Table 1. We also clarified the RE model is a “county-level RE model” in the footnote of the table.

Comment 8: You used three-year averages to test the parallel trend assumption. Since your analysis is conducted using annual data, a more convincing approach is to do an event analysis using annual data, and see whether the coefficients of being treated are significant before 2010. Another “standard” approach is to graphically present the average values of the outcome variables for the treated and control groups, or a residualized plot with some minimal set of control variables.

Response: Following the reviewer’s suggestion, we conducted an event study to perform the parallel trend assumption test. We added a new equation (2) and a description for this test at the end of “*the DID approach*” subsection (right before “*the FE model*” subsection)

$$Y_{it} = \alpha_i + \sum_{k=-5}^{-1} \beta_0 \mathbf{1}(k = t) + \sum_{m=1}^5 \beta_1 \mathbf{1}(m = t) + \mathbf{X}'_{it}\boldsymbol{\gamma} + \delta_t + \varepsilon_{it}. \quad (2)$$

The revised text now reads:

“To test the parallel assumption, we formally estimate the following event study specification:

$$Y_{it} = \alpha_i + \sum_{k=-5}^{-1} \beta_0 \mathbf{1}(k = t) + \sum_{m=1}^5 \beta_1 \mathbf{1}(m = t) + \mathbf{X}'_{it}\boldsymbol{\gamma} + \delta_t + \varepsilon_{it}, \quad (2)$$

where the β_0 terms are the coefficients on the dummy variables for each of the pre-GECP years (2006-2010) and the β_1 terms are the coefficients on the dummy variables for each year after GECP implementation (2011-2015). The reference year is 2011. These coefficients thus measure changes in grassland quality relative to grassland quality in 2011.”

The results of the event study are presented in Figure S1, in place of the old table (Table S2). The results show that in all years before GECP implementation, the coefficients are not statistically different from zero, while in years after the GECP started, the impacts are statistically significant at <5% level. Specifically, the coefficients in the years after GECP implementation range from 3.8% to 5.7%. This suggests that parallel trend assumption is plausible, which are consistent with our previous findings.

Figure S1. Impacts of the GECP on grassland quality at the county level: event study analysis
 Note: The hollow symbol denotes statistically insignificant estimates at the 10% level. The solid symbol denotes statistically significant estimates at <5% level.

Comment 9: I had a hard time understanding how you conducted the common trend test in Table S5 without seeing an estimation equation. For example, in column (1), how does “payment intensity” vary across years? Suppose you used $PaymentIntensity_{it}$, then is the intensity equal to zero before 2010? Or you used a certain average of the payment intensity? Again, you could simply conduct an event analysis to establish common trend.

Response: We have added a formal specification of the pre-trend test for the FE model at the household level in our Methods section. The change in the manuscript is reflected here:

“To rule out reverse causality from the outcome variables to our key independent variable (payment intensity), we conduct an event study using the following specification:

$$Y_{ijklt} = \alpha_{ijkl} + \sum_{m=-3}^{-1} \beta_0 (\bar{P}_{ijklT} \times D_m) + \sum_{n=5}^7 \beta_1 (\bar{P}_{ijklT} \times D_n) + \mathbf{H}'_{ijklt} \boldsymbol{\gamma} + \mathbf{V}'_{jklT} \boldsymbol{\delta} + \mathbf{T}'_{klt} \boldsymbol{\theta} + \mathbf{C}'_{lt} \boldsymbol{\vartheta} + \mu_{ijkl} + \tau_t + \varepsilon_{ijklt}, \quad (4)$$

where the β_0 terms are the coefficients on a set of interaction terms, defined as the dummy variables for each of the pre-GECP years (2008-2010) (D_m) multiplied by the average payment intensity during 2015-2017 (\bar{P}_{ijklT}). Similarly, the β_1 terms are the coefficients on a set of interaction terms, defined as the dummy variables for each year after the GECP (2015-2017) (D_n) multiplied by the average payment intensity during 2015-2017 (\bar{P}_{ijklT}). The reference year is thus 2010. Given this setup, a small

and statistically insignificant estimate of β_0 will lend support to the plausibility of the pre-trend assumption.”

The results show that the estimates of β_0 are all statistically insignificant at 10% level, which indicates that the reverse causality from outcome variables to the key independent variable does not exist.

Comment 10: You provided convincing evidence that richer households received more GECP payments. But why? What are the criteria used in deciding who receives how much? You should include in the paper at least a couple of paragraphs describing in more details important program design features of GECP, so that one will have an easier time to understand your estimation results.

Response: The way the GECP program is designed, the size of GECP payments are made relative to the amount of land a herder has. To be more specific, the total GECP payment received by herders is calculated by multiplying the area (ha) by the regulatory standard (yuan/ha). We will explain the reasons why richer households benefit from this rule on the basis of both land area and regulatory standards.

Regarding land area: Because households with more land received more GECP payments (given the regulatory standards of the program), and richer households tend to have more land, it follows logically that richer households tended to receive more GECP payments through this system.

Regarding the regulatory standards: While China’s central government has set up a uniform national standard as a guideline for all counties in all provinces, individual provinces are able to adjust these regulatory standards based on the average grassland quality in each county and set up county-specific standards. The adjustment rule used by provincial governments in determining county-specific standards is that the higher (average) grassland quality a county has, the higher the regulatory standard, because the opportunity cost for giving up raising animals on higher quality grassland is higher. Since richer households usually have higher quality grassland, they receive more GECP payments per unit of grassland.

We agree with you that such descriptions will help readers have a better understanding of the estimation results. We thus added a subsection at the end of the 5th paragraph in the Introduction on page 4. The added sentences are quoted here:

“A household’s total GECP income depends on the acreage on their grassland certificate, as total GECP payment is calculated by multiplying the total certified area managed by a household by a regulatory standard (yuan/ha). This implies that the more certified grassland a herder has, the larger the GECP payment the herder receives. The central government sets a uniform national payment standard, but each province is allowed to set county-specific payment standards according to its historical record of grassland quality. A county with high baseline

grassland quality will be assigned a higher payment standard to compensate for higher levels of lost income from the livestock sector due to GECP implementation.”

Comment 11: Since GECP is designed to alter herder behavior, and since you do observe behavioral changes (e.g., reduced stock of sheep), one would expect that non-program income (i.e., total household income minus GECP payments) will change. Theoretically the non-program income is likely to decrease (that’s why external payment is needed to induce non-voluntary behavioral changes), but a well-designed program in fact can lead to income increases. I strongly suggest that, in addition to looking at the program’s effects on total income, you also look at the program’s effects on non-program income.

Response: Following the reviewer’s suggestion, we have added these results in Col. (4) of Table 2. Non-program income consists of two parts: pastoral income (Col. 2) and non-pastoral income (Col. 3). As demonstrated in Cols. (2) and (3), the GECP has an insignificant (but positive) impact on pastoral income, while it has a positive and significant impact on non-pastoral income. However, as shown in Figure 2, non-pastoral income only accounts for a small portion of total non-program income. Our results therefore show that non-program income does not change significantly due to the GECP (Table 2, Col. 4).

We do observe behavioral changes (e.g. reduced stock of sheep) among herders. However, this does not necessarily contradict our earlier results about unchanged non-program income (including net pastoral income and non-pastoral income). One possibility is that while farmers reduced sheep stock, the price of sheep increased. Unfortunately, we do not have enough price data to test this hypothesis.

We have incorporated these points in the revised manuscript. The revised section reads:

“We also used a household-level FE model to estimate the impact of GECP payments on total household income, net pastoral income, non-pastoral income, and non-program income. Non-program income includes net pastoral income and non-pastoral income. Table 2 shows that GECP significantly raised both total household income and non-pastoral income of herders but had little effect on their pastoral income and non-program income. The FE models estimated that a 10% increase in annual GECP payments led to an increase of 2.54% in total household income per capita (Column 1) and a 1.67% increase in non-pastoral income per capita (Column 3).”

Table 2. Estimated impacts of the GECP on herder income: household level, FE model

	(1)	(2)	(3)	(4)
Dependent variable	Household income per capita	Net pastoral income per capita	Non-pastoral income per capita	Non-GECP income per capita
Annual GECP payment (yuan)	0.254*** (0.096)	0.062 (0.114)	0.167** (0.078)	0.056 (0.095)
Control variables	Yes	Yes	Yes	Yes
Household fixed effect	Yes	Yes	Yes	Yes
Year fixed effect	Yes	Yes	Yes	Yes
No. of observations	2,024	2,024	2,024	2,024
R^2	0.656	0.679	0.828	0.629

Note: This table provides the results from the FE model using household-level data (Equation (3)). All dependent variables and the key independent variable, annual GECP payment, are taken as an inverse hyperbolic sine transformation to define zero numbers, following $\ln(y+(y^2+1)^{1/2})$. Household income includes net pastoral income, non-pastoral income, and GECP payment. Both household and year fixed effects are controlled for. Household-, village-, and township-level time-variant variables also are controlled for. Household-level controls include quantity of labor used in raising livestock, operated farm size, share of joint operated area, total number of different plots, a dummy variable for grassland harvesting, and a dummy variable for planting crop/fodder. Village-level controls include an indicator of whether a village has local grassroots measures in place to limit grazing intensity and an indicator whether a village has a formal government-run monitoring system. Township-level controls include farm-gate livestock prices, hay prices, wages for non-pastoral employment, and grassland rental prices. County-level climate variables include cumulative rainfall and mean temperature from May to October in each year. Standard errors are in parentheses, clustered by village and year. The results are consistent between an inverse hyperbolic sine transformation and a log transformation of the dependent variable in each column. The log transformation results are available upon request.

* $p < 0.10$, ** $p < 0.05$, *** $p < 0.01$

Comment 12: One effect you identified is that herders rented more land in response to the program. Where did the rental land come from? Is it grassland from other herders? Or from other kinds of land converted to grassland? Did the program lead to more or less grassland in total? These questions are important in understanding the overall ecological impacts of GECP.

Response: We agree with the reviewer that it is important to analyze the GECP impact on total grassland area. Unfortunately, we do not have such data at either the county level nor the household level. Nevertheless, as part of our extensive interviews with herders (For example, asking them from whom they rented the land, details in the rental contract, rental fee, etc.), we have a very concrete and detailed picture of the grassland rental market in the study area (where such markets exist). Most land rental transactions occurred within the same village (82% within village, 12% within township, and 6% outside township). Rental periods are usually short—over 75% of rental land has a rental period of 1 year or less than 1 year. While we might not know about the total amount of grassland area, we do know from our interview questions that about 90% rental land in the sample is rented from other herders (whom may or may not have been surveyed in our sample) and 10% from the public land owned by the village.

To make this point clearer, we revised the Results section to highlight the fact that the rented in land in our sample comes from other herders. The revised section reads as follows:

“When looking at the household level, the analysis also demonstrates that herders enlarged operated grassland area in response to the GECP program. Table 5, Column 5 shows that a 10% increase in payment per hectare leads to a 0.29% increase in the likelihood of renting in grassland. The effect, which is statistically significant, suggested that there exists a grassland rental market between herders, although the small magnitude of the estimate suggests that this market may be incomplete or underdeveloped. Unfortunately, data limitations hinder our ability to further examine the impacts of the GECP program on total grassland area—similar to our analysis of supplementary feeding, we lack county-level data to compare with our household-level results.”

Comment 13: You should mention early on that the treatment and control counties are border counties in the 2011-2015 and post-2016 provinces. Currently these are mentioned in the Data sub-section of the Methods section.

Response: We thank the reviewer for this reminder. We have moved these descriptions of these provinces to the earlier Results section to better inform the reader and be consistent with journal formatting. The revised section on page 7 reads as follows:

“Using a county-level DID method, which better controls for confounding factors, we found that GECP’s impact on grassland quality is positive and statistically significant but small in magnitude. In this DID setup, the treatment group includes all counties in the five North and Northwestern program provinces that were covered in the GECP-I in 2011–2015 (i.e. Xinjiang, Qinghai, Gansu, Ningxia, and

Inner Mongolia). The control group includes all counties in the five North and Northeastern provinces that were not covered by GECP-I until 2016 (i.e. Shanxi, Hebei, Liaoning, Jilin, and Heilongjiang). The pre-program period is 2008–2010, and the post-program period is 2011–2013. The implementation of GECP leads to a 3.2% increase in NDVI (Table 1, Column 1). We tested the parallel-trend assumption using an event study analysis (Figure S1) and believe that the assumption is valid. We also employed numerous robustness checks, such as selecting different time periods and different treatment and control groups for analysis (Tables S2 and S3). The results remained robust when using a random-effects (RE) model (Table 1, Column 2). These checks confirmed our main finding that GECP leads to positive, but small, impacts. We also found that GECP started to have positive impacts on improving grassland quality about two years after program implementation (Table S3).”

Additionally, the definitions of treatment counties are also provided in the footnote of Table 1.

Comment 14: Readers of the DID literature often are interested in the comparisons of the summary statistics between the treated and control groups. Table S6 are about the provinces of the different phases, which include not only the provincial border counties. You should extend this table or have a separate table that shows the comparisons.

Response: We thank the reviewer for this suggestion. We have included the comparisons of the summary statistics between the treated and control groups for different subsamples (Tables S5-S7) in the supplementary materials, and have attached them here for review.

Table S5. Summary statistics for control variables in all counties, separately for control and treatment groups

Variable	All counties					
	Control group (5 N-NE provinces)			Treatment group (5 N-NW provinces)		
	Obs.	Mean	Std. Dev.	Obs.	Mean	Std. Dev.
Rainfall (mm):						
May 2010	279	68.4	32.0	286	47.5	31.1
June 2010	279	36.5	21.6	286	36.0	29.5
July 2010	279	136.7	79.8	286	57.4	52.4
August 2010	279	177.7	80.3	286	54.4	46.1
September 2010	279	64.0	33.7	286	43.4	30.5
October 2010	279	31.3	17.6	286	24.8	15.0
Average temp. (□)						
May 2010	279	16.3	2.5	286	13.9	3.9
June 2010	279	22.6	1.9	286	19.0	4.7
July 2010	279	24.1	2.3	286	21.6	4.3
August 2010	279	21.8	2.2	286	19.6	4.1
September 2010	279	16.8	2.4	286	14.7	3.5
October 2010	279	8.5	3.3	286	7.2	3.9
PSDI:						
May 2010	279	1.9	2.3	286	1.7	3.3
June 2010	279	-0.7	1.7	286	0.9	4.1
July 2010	279	-1.2	2.5	286	0.0	4.5
August 2010	279	-0.1	2.9	286	-0.6	4.4
September 2010	279	0.0	2.8	286	0.0	4.5
October 2010	279	0.1	2.8	286	0.8	4.5
Per-capita county fiscal income (yuan)	279	1221.6	1013.9	286	1806.7	3510.4

Table S6. Summary statistics for control variables in all counties in border provinces, separately for control and treatment groups

Variable	All counties in border provinces					
	Control group (5 N-NE provinces)			Treatment group (Inner Mongolia)		
	Obs.	Mean	Std. Dev.	Obs.	Mean	Std. Dev.
Rainfall (mm):						
May 2010	279	68.4	32.0	83	59.5	28.5
June 2010	279	36.5	21.6	83	23.6	27.0
July 2010	279	136.7	79.8	83	60.6	53.7
August 2010	279	177.7	80.3	83	66.5	36.5
September 2010	279	64.0	33.7	83	54.4	34.5
October 2010	279	31.3	17.6	83	30.0	19.8
Average temp. (°C)						
May 2010	279	16.3	2.5	83	14.0	2.1
June 2010	279	22.6	1.9	83	21.2	1.9
July 2010	279	24.1	2.3	83	23.4	2.3
August 2010	279	21.8	2.2	83	19.5	2.5
September 2010	279	16.8	2.4	83	14.5	2.5
October 2010	279	8.5	3.3	83	5.4	3.3
PSDI:						
May 2010	279	1.9	2.3	83	2.5	1.8
June 2010	279	-0.7	1.7	83	-1.0	1.5
July 2010	279	-1.2	2.5	83	-2.9	1.8
August 2010	279	-0.1	2.9	83	-3.4	2.1
September 2010	279	0.0	2.8	83	-1.6	3.0
October 2010	279	0.1	2.8	83	0.0	2.8
Per-capita county fiscal income (yuan)	279	1221.6	1013.9	83	3352.5	5231.0

Table S7. Summary statistics for control variables in border counties, separately for control and treatment groups

Variable	Border counties					
	Control group (5 N-NE provinces)			Treatment group (Inner Mongolia)		
	Obs.	Mean	Std. Dev.	Obs.	Mean	Std. Dev.
Rainfall (mm):						
May 2010	35	70.6	26.3	31	66.9	21.2
June 2010	35	28.8	9.8	31	23.4	8.3
July 2010	35	128.3	78.1	31	95.2	59.6
August 2010	35	118.4	50.0	31	91.7	33.2
September 2010	35	63.1	39.3	31	55.0	38.0
October 2010	35	41.1	19.2	31	35.9	18.4
Average temp. (°C)						
May 2010	35	14.8	1.8	31	13.5	2.0
June 2010	35	21.6	2.2	31	20.8	2.2
July 2010	35	23.2	1.5	31	22.3	1.9
August 2010	35	20.2	2.1	31	19.0	2.4
September 2010	35	15.2	2.2	31	13.9	2.5
October 2010	35	6.0	2.5	31	4.5	2.8
PSDI:						
May 2010	35	2.5	2.3	31	2.5	2.1
June 2010	35	-0.9	1.9	31	-1.4	1.3
July 2010	35	-1.3	2.9	31	-2.5	2.2
August 2010	35	-1.2	3.5	31	-2.7	2.6
September 2010	35	0.1	3.2	31	-1.0	3.1
October 2010	35	1.5	3.1	31	0.9	2.8
Per-capita county fiscal income (yuan)	35	886.9	671.6	31	1372.0	3288.1

Reviewer comments, second round -

Reviewers' comments:

Reviewer #1 (Remarks to the Author):

The revised paper had a big improvement. But there are still two questions confused me. (1) I also confirmed the data in Inner Mongolia or grasslands in other part should be added, because this is a study on the whole grasslands in China. And the grasslands in the western part and eastern part are extremely different. (2) Some results should be quantified. The author did many modelling and made many tables but lack spatial analysis. For a large scale study, a quantified results on spatial heterogeneous are meanful. Figure 1 clearly showed the spatial heterogeneity, so the author should use more words and numbers to decript the difference on difference places or grassland types. There are similar problems with other factos.

Reviewer #2 (Remarks to the Author):

The revision has successfully addressed my concerns and I have no more comments. This is a very nice paper!

Response to Reviewer #1

Comment 1: The revised paper had a big improvement. But there are still two questions confused me. (1) I also confirmed the data in Inner Mongolia or grasslands in other part should be added, because this is a study on the whole grasslands in China. And the grasslands in the western part and eastern part are extremely different.

Response to Comment 1: We would like to thank the reviewer again for the valuable comments. During the first round of revisions, COVID-19 travel restrictions prevented us from returning to the field to collect additional data in Inner Mongolia. Fortunately, we were allowed to travel during September and October 2020. We therefore added data from not only Inner Mongolia but also Xinjiang and Tibet. Figure S3 shows the current study area and sample distribution.

With the new data (the number of observations rose from 2,024 to 3,110), we updated all of the tables and text related to household-level analysis. We summarize the major changes below for your convenience. Please find the full changes in the tracked-changes manuscript.

First, we updated the estimated impact of the GECP program on grassland quality in Column 3 of Table 1. Even with the additional data, the coefficient, although re-estimated, changed from only 0.010* to 0.011*.

Table 1. Estimated impact of the GECP on grassland quality: county and household levels

Models	(1)	(2)	(3)
	County level		Household level
	DID	RE	FE
$P \times T$	0.032*** (0.005)	0.024*** (0.005)	
$\log(\text{payment intensity})$ (yuan/ha)			0.011* (0.006)
No. of observations	3,425	3,425	3110
No. of counties/households	574	574	821
R^2	0.989	0.806	0.958

Note: The dependent variable is NDVI in log form. Column (1) provides the results from the DID approach using county-level data (Equation (1)). The treatment group ($P = 1$) includes the counties in five North and Northwestern program provinces that were covered in GECP-I, i.e., Xinjiang, Qinghai, Gansu, Ningxia, and Inner Mongolia. The control group ($P = 0$) includes the counties in five North and Northeastern provinces that were not covered by GECP-I until 2016, i.e., Shanxi, Hebei, Liaoning, Jilin, and Heilongjiang. The pre-program period ($T = 0$) is 2008–2010. The post-program period ($T = 1$) is 2011–2013. Column (2) presents the results from the county-level random-effect model. In Columns (1) and (2), year and province fixed effects are controlled for. Climate controls include monthly rainfall, temperature, and PSDI for May–October. Socioeconomic control includes per-capita county fiscal income. Robust standard errors are in parentheses, clustered at the county level. Column (3) reports the results from the household-level FE model (Equation (3)). Both household and year fixed effects are controlled for. Household-, village-, and township-level time-variant variables also are controlled for.

Household-level controls include quantity of labor used in raising livestock, operated farm size, share of joint operated area, total number of different plots, a dummy variable for grassland harvesting, and a dummy variable for planting crop/fodder. Village-level controls include an indicator of whether a village has local grassroots measures in place to limit grazing intensity, an indicator of whether a village has a formal government-run monitoring system and climate variables (cumulative rainfall and mean temperature from May to October in each year). Township-level controls include farm-gate livestock prices, hay prices, wages for non-pastoral employment, and grassland rental prices. Standard errors are in parentheses, clustered by village and year.

* $p < 0.10$, ** $p < 0.05$, *** $p < 0.01$

The text in the revised manuscript reads as follows:

Table 1, Column 3, shows that a 10% increase in payment intensity led to an increase of 0.11% in NDVI. In other words, if payment intensity doubles, NDVI would increase by only 1.1%.

Second, we updated the estimated impacts on herder income in Table 2, using the new data from all five provinces (Table 2, Panel A).

Table 2. Estimated impacts of the GECP on herder income: household level, FE model

	(1)	(2)	(3)	(4)
Panel A. Overall impacts on income				
Dependent variable	Household income per capita	Net pastoral income per capita	Non-pastoral income per capita	Non-GECP income per capita
Annual GECP payment (yuan)	0.366*** (0.129)	0.130 (0.147)	0.048 (0.053)	0.144 (0.149)
Control variables	Yes	Yes	Yes	Yes
Household fixed effect	Yes	Yes	Yes	Yes
Year fixed effect	Yes	Yes	Yes	Yes
No. of observations	3,474	3,474	3,474	3,474
R^2	0.731	0.749	0.863	0.744
Panel B. Heterogeneous impacts by grassland types				
Grassland	0.406* (0.206)	0.233 (0.234)	0.028 (0.080)	0.188 (0.235)
Meadow	0.235* (0.121)	-0.233 (0.161)	0.118** (0.055)	-0.039 (0.152)
Desert	0.268 (0.335)	0.305 (0.318)	0.005 (0.095)	0.287 (0.316)
Panel C. Heterogeneous impacts by socioeconomic variables				
Education of the labors (years)	-0.004 (0.003)	-0.004 (0.004)	0.021*** (0.003)	-0.001 (0.003)
Distance to the closest township-level road (km)	-0.000 (0.000)	-0.000 (0.000)	-0.001** (0.000)	-0.000 (0.000)
Grassland area per capita (hundred ha in log)	0.007 (0.012)	0.012 (0.016)	-0.033*** (0.009)	0.009 (0.014)

Note: Panel A provides the estimated overall impacts on income from the FE model using household-level data (Equation (3)). All dependent variables and the key independent variable, annual GECP payment, are taken as an inverse hyperbolic sine transformation to define zero numbers, following $\ln(y+(y^2+1)^{1/2})$. Household income includes net pastoral income, non-pastoral income, and GECP payment. Both household and year fixed effects are controlled for. Household-, village-, and township-level time-variant variables also are controlled for. Household-level controls include quantity of labor used in raising livestock, operated farm size, share of joint operated area, total number of different plots, a dummy variable for grassland harvesting, and a dummy variable for planting crop/fodder. Village-level controls include an indicator of whether a village has local grassroots measures in place to limit grazing intensity, an indicator whether a village has a formal government-run monitoring system and climate variables (cumulative rainfall and mean temperature from May to October in each year). Township-level controls include farm-gate livestock prices, hay prices, wages for non-pastoral employment, and grassland rental prices. Standard errors are in parentheses, clustered by village and year. The results are consistent between an inverse hyperbolic sine transformation and a log transformation of the dependent variable in each column. The log transformation results are available upon request. Panel B provides the heterogeneous impacts on income by grassland type. We use the grassland type with the largest area in a county as its major grassland type. Each set of coefficient and corresponding standard error are from one single regression model. All model specifications are the same as in Panel A. Panel C provides the heterogeneous impacts on income by different socioeconomic variables.

* $p < 0.10$, ** $p < 0.05$, *** $p < 0.01$

After adding data from the three new provinces, the results are largely consistent with those in the original manuscript. The corresponding text reads as follows:

We also used a household-level FE model to estimate the impact of GECP payments on total household income, net pastoral income, non-pastoral income, and non-program income. Non-program income includes net pastoral income and non-pastoral income. Panel A of Table 2 shows that the GECP program significantly raised both total household income but had little effect on pastoral income, non-pastoral income and non-program income. The FE models estimated that a 10% increase in annual GECP payments led to an increase of 3.66% in total household income per capita (Column 1). Although the coefficients of pastoral income (Column 2), non-pastoral income (Column 3), and non-program income (Column 4) are statistically insignificant, they are all positive. This indicates that, although the GECP program has an overall impact on herder income, in general, the program did not boost any specific types of income. This may be due to the fact that there are differences in the emphasis on different types of specific sources of income in the different parts of the sample.

Third, we updated the estimated impacts of the GECP payment on income equity in Table 3, using five-province data.

Table 3. Estimated impacts of the GECP payment on income equity: household level, FE model

	(1)	(2)	(3)
Dependent variable	Household income per capita	Net pastoral income per capita	Non-pastoral income per capita
Low-income group	0.461*** (0.135)	0.112 (0.150)	-0.062 (0.054)
Middle-income group	0.361*** (0.136)	0.125 (0.153)	0.029 (0.052)
High-income group	0.332** (0.131)	0.137 (0.146)	0.091** (0.046)
No. of observations	3,469	3,469	3,469
R^2	0.732	0.749	0.865

Note: This table provides the results from the FE model using household level data (Equation (6)). All dependent variables and the key independent variable, annual GECP payment, are taken as an inverse hyperbolic sine transformation to define zero numbers, following $\ln(y+(y^2+1)^{1/2})$. Household income includes net pastoral income, non-pastoral income, and GECP payment. Both household and year fixed effects are controlled for. Household-, village-, and township-level time-variant variables also are controlled for. Household-level controls include quantity of labor used in raising livestock, operated farm size, share of joint operated area, total number of different plots, a dummy variable for grassland harvesting, and a dummy variable for planting crop/fodder. Village-level controls include an indicator of whether a village has local grassroots measures in place to limit grazing intensity and an indicator of whether a village has a formal government-run monitoring system. Township-level controls include farm-gate livestock prices, hay prices, wages for non-pastoral employment, and grassland rental prices. Village-level climate variables include cumulative rainfall and mean temperature from May to October in each year. Standard errors are in parentheses, clustered by village and year.

* $p < 0.10$, ** $p < 0.05$, *** $p < 0.01$

Again, the results that use the expanded data set are similar to those in the original manuscript. The text in the new version reads as follows:

Our FE estimates confirm this trend of the widening of income inequality. A 1% increase in annual household GECP payments led to an increase of 0.46% in household income for the low-income group; 0.36%, for the medium-income group; and 0.33%, for the high-income group (Table 3, Column 1). These estimates imply that doubling GECP payments increased household income

per capita by 407 RMB for the low-income group, 1,470 RMB for the medium-income group, and 6,955 RMB for the high-income group. Even though GECP payments represented a smaller portion of total income for the high-income group, in absolute terms, high-income households received more money from the program.

An examination of the subcategories of income reveals that, although GECP payments had virtually no effect on any income group (Table 3, Column 2), non-pastoral income increased for the high-income group (Table 3, Column 3) but only by a small amount. A 1% increase in total payments led to a non-pastoral income increase of 0.09% for the high-income group. In other words, doubling the total payments would have resulted in an annual rise in non-pastoral income of 180 RMB per capita for the high-income group. Given that average herder income before the program was already 9,100 RMB per capita, these are not substantially large increases.

Fourth, in the revised manuscript, we also updated the mechanism analysis, using data from all five provinces. We then updated Table 5 and the corresponding text, as seen below:

The analysis at the county level shows that herders reduced sheep inventory but did not reduce cattle inventory as a response to the GECP program. Applying the DID approach to county-level livestock data, we found that the GECP program reduced year-end sheep inventory by 12.1%, but the effect on cattle inventory was statistically insignificant (Table 4). Estimates derived from household livestock data show that an increase in the per-hectare GECP payment had no significant effect on cattle inventories (Table 5, Columns 1–3). This finding indicates that increasing the level of payment does not appear to reduce livestock inventories, given the current approach to implementing the GECP program (that is, without augmenting the current program with other measures).

Table 5, Column 4 shows that GECP payments were significantly correlated with supplementary feeding at the household level. The results indicate that a 10% increase in payment per hectare leads to a 0.75% increase in supplementary feeding. This suggests that herders increased supplementary feeding as a response to GECP payments but did so at only a low level. Unfortunately, we cannot confirm this at the county level, as county-level data on supplementary feeding do not currently exist.

When considering the household level, the analysis also demonstrates that herders enlarged their operated grassland area in response to the GECP program. Table 5, Column 5 shows that a 10% increase in payment per hectare leads to a 0.12% increase in the likelihood of renting in grassland. The effect,

which is statistically significant, suggests that there is a grassland rental market between herders, although the small magnitude of the estimate suggests that this market may be incomplete or underdeveloped. Unfortunately, data limitations hinder our ability to further examine the impacts of the GECP program on total grassland area; similar to our analysis of supplementary feeding, we lack county-level data to compare with our household-level results.

Table 5. Estimated impacts of the GECP on herder behavior: household level

	(1)	(2)	(3)	(4)	(5)
Dependent variable	Livestock inventory	Cattle inventory	Sheep inventory	Supplementary feeding	Grassland rent in
$\log(\text{payment intensity (yuan/ha)})$	-0.003	-0.032	0.021	0.075**	0.012**
	-0.025	-0.035	-0.033	-0.033	-0.005
No. of observations	3,429	3,429	3,429	3,473	3,474
R^2	0.723	0.898	0.938	0.826	0.892

Note: This table reports the results from the household-level FE model (Equation (3)). All dependent variables are taken as a hyperbolic sine transformation to define zero numbers, following $\ln(y+(y^2+1)^{1/2})$, except the variable that indicates grassland rent in or not. The payment intensity is taken in log form transformation. Both household and year fixed effects are controlled for. Household-, village-, and township-level time-variant variables also are controlled for. Household-level controls include quantity of labor used in raising livestock, operated farm size, share of joint operated area, total number of different plots, a dummy variable for grassland harvesting, and a dummy variable for planting crop/fodder. Village-level controls include a variable that indicates whether a village has local grassroots measures in place to limit grazing intensity, a variable that indicates whether a village has a formal government-run monitoring system and climate variables (cumulative rainfall and mean temperature from May to October in each year). Township-level controls include farm-gate livestock prices, hay prices, wages for non-pastoral employment, and grassland rental prices. Standard errors are in parentheses, clustered by village and year.

* $p < 0.10$, ** $p < 0.05$, *** $p < 0.01$

Comment 2: (2) Some results should be quantified. The author did many modelling and made many tables but lack spatial analysis. For a large-scale study, a quantified results on spatial heterogeneous are meaningful. Figure 1 clearly showed the spatial heterogeneity, so the author should use more words and numbers to depict the difference on difference places or grassland types. There are similar problems with other factors.

Response to Comment 2: We thank the reviewer for pointing out the heterogeneity issue again. We agree that this is a very important issue, as our study area is so wide. In the first round of the revisions, in fact, we did add a

section on heterogeneous impacts in which we considered the impact of the program in counties that have different levels of grassland quality (using grassland types as a measure of spatial heterogeneity). In looking at that version of the manuscript (R1), it seems that we probably did not highlight this new part of our analysis sufficiently. In this version of the manuscript, we have a subsection titled “Heterogeneity analysis” (see pages 12 and 16 in the newly revised manuscript).

Below, we summarize the spatial heterogeneity analysis used in our study for your review.

First, the (spatial) heterogeneous impacts on grassland quality:

To analyze spatial heterogeneity at the county level, we used grassland type as an indicator under the assumption that different grassland types reflect different natural resource endowments and affect the nature of each region’s economic activities. Following Ma and Xu (2019), we adopted the classification of five grassland types (i.e., meadow, grassland, desert, shrubland, and herbosa). Our county-level dataset covers all five types (see Figure R1 below).

Figure R1. Map of grassland types in study area

After analyzing the spatially heterogeneous impacts of the GECP program on grassland quality at the county level, we added Panel C to Table 6 in the revised manuscript. According to the newly added results, GECP impacts on improving grassland quality are positive and statistically significant in four out of the five grassland types (grassland, desert, shrubland, and herbosa but not meadows). In other words, we

found that the GECP program raised grassland quality in all areas of China (including counties that are classified as having primarily grassland, desert, shrubland, and herbosa), with the exception of counties that are classified as having primarily meadows. The coefficients of the GECP program variable vary across the four areas for which statistically significant impacts were found (from 0.018 for shrubland to 0.055 for grassland). The coefficient on the GECP program variable for counties primarily with meadows also is positive, although the point estimate is small (and, as stated, is statistically insignificant). The results suggest that, even though there may be some spatial heterogeneity based on the natural resource endowment of counties, the results are mainly consistent across space: The GECP program has had a positive effect on grassland quality in most counties (classified as having different types of grassland).

In the revised manuscript, we now have added a paragraph in the Results section (pages 13–14), which reads as follows:

Third, we explored potential spatial heterogeneity as related to the impact of the GECP program. We use grassland type as an indicator for spatial heterogeneity, under the assumption that different types of grassland reflect different natural resource endowments and affect the nature of each region's economic activities. Following Ma and Xu (2019), we adopt the classification of five grassland types (i.e., meadow, grassland, desert, shrubland, and herbosa), which are all covered in our county-level data.

The results (Table 6, Panel C) suggest that, although there is some spatial heterogeneity based on the nature of the resource endowments of counties, the effects are consistent mainly across space: The GECP program has had a positive effect on grassland quality in most counties (classified as having different types of grassland) in the program area. More specifically, the GECP program raised grassland quality in all counties (including counties classified as having primarily grassland, desert, shrubland, and herbosa), with the exception of counties that are classified as having primarily meadows. The coefficients for the GECP program variable vary across the four areas in which statistically significant impacts were found (from 0.018 for shrubland to 0.055 for grassland). The coefficient for the GECP program variable for counties with meadows also is positive, although the point estimate is small (and, as stated, statistically insignificant).

Second, the heterogeneous impacts on herder income:

We also added the (spatial) heterogeneous impacts on herder income in Panel B of Table 2 and the heterogeneous impacts by socioeconomic variables in Panel C.

Table 2. Estimated impacts of the GECP on herder income: household level, FE model

	(1)	(2)	(3)	(4)
Panel A. Overall impacts on income				
	Household income per capita	Net pastoral income per capita	Non-pastoral income per capita	Non-GECP income per capita
Dependent variable				
Annual GECP payment (yuan)	0.366*** (0.129)	0.130 (0.147)	0.048 (0.053)	0.144 (0.149)
Control variables	Yes	Yes	Yes	Yes
Household fixed effect	Yes	Yes	Yes	Yes
Year fixed effect	Yes	Yes	Yes	Yes
No. of observations	3,474	3,474	3,474	3,474
R ²	0.731	0.749	0.863	0.744
Panel B. Heterogeneous impacts by grassland types				
Grassland	0.406* (0.206)	0.233 (0.234)	0.028 (0.080)	0.188 (0.235)
Meadow	0.235* (0.121)	-0.233 (0.161)	0.118** (0.055)	-0.039 (0.152)
Desert	0.268 (0.335)	0.305 (0.318)	0.005 (0.095)	0.287 (0.316)
Panel C. Heterogeneous impacts by socioeconomic variables				
Labor average education (years)	-0.004 (0.003)	-0.004 (0.004)	0.021*** (0.003)	-0.001 (0.003)
Distance to the closest township-level road (km)	-0.000 (0.000)	-0.000 (0.000)	-0.001** (0.000)	-0.000 (0.000)
Grassland area per capita (hundred ha in log)	0.007 (0.012)	0.012 (0.016)	-0.033*** (0.009)	0.009 (0.014)

Note: Panel A provides the estimated overall impacts on income from the FE model using household-level data (Equation (3)). All dependent variables and the key independent variable, annual GECP payment, are taken as an inverse hyperbolic sine transformation to define zero numbers, following $\ln(y+(y^2+1)^{1/2})$. Household income includes net pastoral income, non-pastoral income, and GECP payment. Both household and year fixed effects are controlled for. Household-, village-, and

township-level time-variant variables also are controlled for. Household-level controls include quantity of labor used in raising livestock, operated farm size, share of joint operated area, total number of different plots, a dummy variable for grassland harvesting, and a dummy variable for planting crop/fodder. Village-level controls include an indicator of whether a village has local grassroots measures in place to limit grazing intensity and an indicator whether a village has a formal government-run monitoring system. Township-level controls include farm-gate livestock prices, hay prices, wages for non-pastoral employment, and grassland rental prices. Village-level climate variables include cumulative rainfall and mean temperature from May to October in each year. Standard errors are in parentheses, clustered by village and year. The results are consistent between an inverse hyperbolic sine transformation and a log transformation of the dependent variable in each column. The log transformation results are available upon request.

Panel B provides the heterogeneous impacts on income by grassland type. We use the grassland type with the largest area in a county as its major grassland type. Each set of coefficient and corresponding standard error are from one single regression model.

All model specifications are the same as in Panel A.

Panel C provides the heterogeneous impacts on income by different socioeconomic variables.

* $p < 0.10$, ** $p < 0.05$, *** $p < 0.01$

In the revised manuscript, we have added a paragraph in the Heterogeneity Analysis section (pages 15–16), which reads as follows:

In addition to the heterogeneous impacts on grassland quality, we also examined the heterogeneous impacts on herder income by different grassland types that reflect spatial heterogeneity (Panel B of Table 2) and certain socioeconomic variables (Panel C of Table 2). We found that the GECP program has different impacts on annual household income per capita in areas with different grassland types and that the impacts on non-pastoral income per capita vary across groups with different socioeconomic characteristics.

First, the GECP program has a larger impact in regard to improving household income per capita in grassland and meadow, whereas the impact in the desert is insignificant (Column 1 of Panel B). Household income per capita has been improved by 40.6% in grassland and 23.5% in meadow regions. This is due to the regulatory standard implemented by the GECP. As noted, regions with high baseline grassland quality, such as grassland and meadow regions, are assigned a higher payment standard to compensate for higher levels of lost income from the livestock sector. Regions with low grassland productivity, such as desert areas, however, are assigned a lower payment standard.

Second, the impacts on non-pastoral income are heterogeneous across groups with different socioeconomic characteristics (Column 3 of Panel C), whereas the impacts on household income, pastoral income,

and non-program income are more homogeneous (Columns 1, 2, and 4 of Panel C). The GECP program has a larger impact on non-pastoral income for herders with higher education, those who live closer to the township-level road, and those with smaller grassland area per capita. This indicates that herders with higher education are probably more competitive in non-pastoral labor markets. Herders who live closer to roads are more likely to obtain a non-pastoral job, as the travel cost is lower and job market information may be more accessible. Herders with fewer grassland areas per capita are more likely to switch to a non-pastoral sector.

Response to Reviewer #2

Comment: The revision has successfully addressed my concerns and I have no more comments. This is a very nice paper!

Response: We thank you again for your insightful comments in the previous round. They were very helpful, and we believe that our manuscript has been greatly improved as a result of the related revisions.

Reviewer comments, third round -

Reviewer #1 (Remarks to the Author):

This revision has successfully addressed my concerns and it has been a very nice paper!